# Effect of Simplified Bonding on Shear Bond Strength between Ceramic Brackets and Dental Zirconia

**DOI:** 10.3390/ma12101640

**Published:** 2019-05-20

**Authors:** Ga-Youn Ju, Soram Oh, Bum-Soon Lim, Hyun-Seung Lee, Shin Hye Chung

**Affiliations:** 1Department of Dental Biomaterials Science, School of Dentistry, Seoul National University, Seoul 03080, Korea; espoir840126@naver.com (G.-Y.J.); hyun11@snu.ac.kr (H.-S.L.); 2Department of Conservative Dentistry, Kyung Hee University Dental Hospital (KHUDH), Seoul 02447, Korea; soram0123@gmail.com; 3Department of Dental Biomaterials Science, School of Dentistry and Dental Research Institute, Seoul National University, Seoul 03080, Korea; nowick@snu.ac.kr

**Keywords:** dental zirconia, orthodontic bracket, 10-MDP, surface treatment, shear bond strength, resin bonding

## Abstract

The aim of this study was to evaluate the long term stability of shear bond strength (SBS) when 10-methacryloyloxydecyl dihydrogen phosphate (10-MDP) containing universal adhesive was used in the ceramic bracket bonding on dental zirconia. Twenty human maxillary incisors were collected. The ceramic bracket was bonded on the buccal enamel surface after the acid-etching and orthodontic primer application (Group CON). Sixty zirconia specimens were sintered, sandblasted and divided into three experimental groups; group CP—ceramic primer followed by an orthodontic primer; group U—universal adhesive; group CU—ceramic primer followed by a universal adhesive. For each specimen, the bracket was bonded onto the treated surface with composite resin (Transbond XT, 3M ESPE). The SBS tested before (CON0, CP0, U0, CU0) and after the artificial aging (CON1, CP1, U1, CU1). The data were statistically analyzed with the Kruskal–Wallis test at a significance level of 0.05. The mean SBS of CON0, CP0, U0 and CU0 were within the clinically acceptable range without significant differences. After the aging process, SBS decreased in all groups. Among the aged groups, CP1 showed the highest SBS. Based on the results, when bonding ceramic brackets to a dental zirconia surface, we can conclude that ceramic primer used with an orthodontic primer, rather than using a universal adhesive, is recommended.

## 1. Introduction

Recently, together with the development of CAD/CAM devices, the use of zirconia in dental restorations are getting more and more popularity. The dental zirconia offers excellent mechanical strength and provides optimal aesthetics compared to porcelain fused to metal restorations [1,2]. The most common type of zirconia used in dentistry is yttrium-stabilized tetragonal zirconia polycrystal (Y-TZP), which remains in the tetragonal phase at room temperature with the addition of 2–3 mol% yttrium [3,4,5]. Crack propagation of Y-TZP is blocked by the transition from the tetragonal to the monoclinic phase [6,7,8]. Furthermore, Y-TZP has the advantage of utilizing CAD/CAM systems [9,10]

Despite superior mechanical properties, zirconia lacks translucency resulting in a white and opaque shade. In restorations in aesthetic regions, translucent porcelain is veneered onto the zirconia-framework to provide a natural appearance. However, chipping, crack and delamination of the veneering porcelain has been indicated as a problem in such restorations. The chipping or crack of the porcelain is attributed to the poor wetting of the veneering, the firing shrinkage of the porcelain and the difference in the thermal expansion coefficient between the zirconia and the porcelain. To overcome those problems, non-veneered zirconia restoration, made of monolithic zirconia (MZ) has been suggested. Since MZ is a single body that has no distinction between upper and lower structures, partial chipping of the restoration rarely occurs. Recently, the MZ has been widely used for single crowns and fixed partial dentures, because it offers excellent mechanical strength as well as suitable aesthetics [3,11].

The number of adult patients performing an orthodontic treatment is increasing for several reasons, not only by the attention paid to aesthetics but also demand for the improvement of the occlusal relationships [12,13]. The use of ceramic brackets is also increasing for the improved aesthetics [13]. When the tooth is restored with MZ, bonding brackets onto the zirconia is a challenging process since the zirconia surface, which has no silica phase, cannot be bonded with composite resin through conventional hydrofluoric acid-etching and silane treatment [11,14,15]. Various bonding protocols have been considered for Y-TZP, including mechanical methods of laser burning, sandblasting and tribochemical treatment to create micro-roughness on the surface [14,15,16,17]. Chemical methods, such as using a bonding agent, chloro-silane vapor or fluorinated plasma, were applied to improve the bond strength between composite resin and Y-TZP [14,18,19,20].

The bonding area of an orthodontic bracket is relatively small compared to that of a regular tooth restoration. Sufficient bond strength is required that is able to endure the shear stress during daily activities such as chewing. In general, when bonding an orthodontic bracket to the enamel surface includes three steps: acid-etching, orthodontic adhesive primer application and orthodontic resin. One of the methods to increase the bond strength between the composite resin and Y-TZP is sandblasting and applying the functional monomer-containing products [5,18,21]. A functional monomer, 10-methacryloyloxydecyl dihydrogen phosphate (10-MDP) is known to improve the bond strength between dental zirconia and composite resin by chemically combining with oxides on the Y-TZP surface [15,22]. When bonding restorations, simplified bonding protocols by using universal adhesives containing functional monomers are used. As such, less complicated, reduced number of steps in bonding procedure would be useful for clinicians to minimize technique sensitivity or unintended errors to achieve stable long-term bond strength in bracket bonding.

The aim of this study was to evaluate the long term stability of shear bond strength (SBS) when 10-MDP-containing universal adhesive was used in the ceramic bracket bonding on dental zirconia.

## 2. Materials and Methods

### 2.1. Preparation of Y-TZP

Sixty cuboidal Y-TZP specimens in 15 (width) × 15 (height) × 3 (thickness) mm were prepared from a green-stage block (LAVA Plus, 3M ESPE, St. Paul, MN, USA) and then the specimens were sintered according to the manufacturer’s instructions.

Each Y-TZP specimen was embedded in polyester resin (EC-304, Aekyung, Seoul, Korea) and its bonding surface was polished with a diamond disc of 500 grit (MD-Piano, Struers, Ballerup, Denmark) under constant water cooling. The samples were immersed in a distilled water with ultrasonic vibration and dried. The surface of Y-TZP was sandblasted with 50 µm alumina (SandStorm Expert, Vaniman, Fallbrook, CA, USA) at a distance of 20 mm and a pressure of 0.4 MPa in the vertical direction for 20 s. The sandblasted Y-TZP were immersed in a distilled water with ultrasonic vibration for 2 min and dried.

### 2.2. Observation of Surface Charateristics of Y-TZP

#### 2.2.1. Examination by Field Emission-Scanning Electron Microscope

The surface topology of Y-TZP before and after sandblasting was examined by using Field emission scanning electron microscope (FE-SEM; S-4700, Hitachi, Tokyo, Japan) at ×500 and ×5000.

#### 2.2.2. Measurement of Surface Roughness

The surface was observed before and after the sandblasting on three samples, each under a confocal laser scanning microscope (CLSM; LSM 800-MAT, Carl Zeiss MicroImaging GmbH, Jena, Germany). The roughness parameters evaluated were average surface roughness (*R*_a_), root mean square roughness (*R*_q_), roughness depth (*R*_z_) and roughness skewness (*R*_sk_) [23].

#### 2.2.3. Measurement of Contact Angle

The contact angle of a water droplet was measured before and after the sandblasting on three samples each by using a contact-angle analyser (Phoenix 150, SEO, Suwon, Korea). The average and standard deviation was recorded after the measurement.

### 2.3. Bonding of Bracket and Shear Bond Strength Test

#### 2.3.1. Experimental Design and Bonding the Bracket to Y-TZP

The experimental design required 60 Y-TZP for three different experimental groups and 20 enamel specimens for a control group. Each group was divided into two subgroups (*n* = 10), according to whether the aging treatment was performed. Figure 1 demonstrates the flow chart of this study and Table 1 shows the specifications of the materials used in this study.

For the specimens in group CON, Sound human maxillary central incisors (*n* = 20) extracted for periodontal reasons were used under a protocol approved by an Institutional Review Board (No. S-D20150041). Maxillary central incisors (*n* = 20) which had been stored in a 0.1% thymol solution at 4 °C for less than 3 months after the extraction, were used. The roots were removed at the cemento-enamel junction using a diamond disk, the crown was fixed to the acrylic resin so that the entire labial surface of the crown would be exposed and parallel to the base. It was cleaned in distilled water with ultrasonic vibration for 2 min. The excessive moisture was dried with air by using three-way syringe. The enamel was etched with 37% phosphoric acid (Scotchbond Universal Etchant, 3M ESPE, Nuess, Germany) for 20 s and then washed for 20 s using water, followed by drying with gentle air from the three-way syringe according to the manufacturers’ instructions. Thereafter, an orthodontic primer (Transbond XT adhesive primer, 3M Unitek, Monrovia, CA, USA) was applied to the enamel in a thin layer and the excessive orthodontic primer was removed by air blowing using three-way syringe according to the manufacturers’ instructions.

For the specimens in group CP, the sandblasted Y-TZP was treated with a ceramic primer (Clearfil ceramic primer, Kuraray, Tokyo, Japan) in a thin layer and dried thoroughly with gentle air of three-way syringe, followed by orthodontic primer application.

For the specimens in group U, the sandblasted Y-TZP was treated with a universal adhesive (Clearfil S^3^ bond, Kuraray, Tokyo, Japan). The adhesive was applied in a thin layer, left for 20 s, dried for 5 s with the air of three-way syringe and light-cured for 10 s.

For the specimens in Group CU, the sandblasted Y-TZP was treated with a ceramic primer, followed by application of a universal adhesive, then light-cured.

A ceramic bracket (Perfect Clear II, Hubit, Uiwang, Korea) for maxillary central incisor (bonding surface area of 12.24 mm^2^) was used. The ceramic bracket used in this study was made of monocrystalline sapphire with retentive beads spreading at the centre of the bracket base. A composite resin (Transbond XT adhesive paste, 3M Unitek, Monrovia, CA, USA) was applied to the bracket base, then the bracket was placed under gentle pressure until the margin of the bracket base reached the surface (i.e., Y-TZP or enamel). Excessive resin was removed with a resin applicator. Light-curing was performed toward the margin of the bracket with a light-emitting diode (LED) curing unit (Elipar Free Light 2, 3M ESPE, St. Paul, MN, USA). Light-cured at each margin for 10 s, therefore light-curing was performed for a total of 40 s. Since the composite resin is located between the bracket and the Y-TZP, a sufficient amount of light was irradiated instead of following the manufacturer’s recommendation (10 s) to prevent any error due to insufficient polymerization of the composite resin.

#### 2.3.2. Shear Bond Strength Test

After bracket bonding, all the samples (*n* = 80) were kept in a 37 °C and relative humidity 100% incubator for 24 h. SBS of forty samples, ten per each group, tested with a universal testing machine (Instron 8848, Instron, Norfolk County, MA, USA) at a crosshead speed of 0.5 mm/min (CON0, CP0, U0 and CU0). The other half were subjected to the aging process by thermocycling for 10,000 cycles at 5 and 55 °C (CON1, CP1, U1 and CU1). The dwelling time in water was 30 s and the transfer time was 20 s. After thermocycling, SBS of the samples tested in the same way. The maximum load-at-failure was calculated in MPa by dividing the maximum load (N) by the area of the bracket base (12.24 mm^2^). The failed surface was examined by FE-SEM at ×30. The amount of residual resin on the surface of each surface was classified according to the adhesive remnant index (ARI) score (Table 2).

Shear bond strength was compared between experimental groups with Kruskal–Wallis test. The Statistical Package for the Social Sciences (Version 22.0; IBM, Armonk, NY, USA) was used, at a significance level of 5%.

## 3. Results

### 3.1. Surface Characteristics of Y-TZP

#### 3.1.1. Field Emission-Scanning Electron Microscopic Examination

When the polished, pre-sandblasted Y-TZP specimen was examined at ×500, a very even surface was found (Figure 2a); when it was examined at ×5000, some directionless scratches were observed (Figure 2b). Examination of the sandblasted Y-TZP specimen at ×500 revealed uneven and roughly dented traces scattered on the surface (Figure 2c). Meanwhile, the ×5000 image shows rougher surface than that of the polished sample (Figure 2d).

#### 3.1.2. Measurement of Surface Roughness

When the polished surface of Y-TZP, not subjected to sandblasting, was examined using CLSM, the Y-TZP appeared relatively even surface (Figure 3a,b); the average *R*_a_ was 0.10 ± 0.00 µm, average *R*_q_ was 0.14 *±* 0.00 µm, average *R*_z_ was 0.96 *±* 0.17 µm and average *R*_sk_ was −1.25 *±* 0.43 µm. The surface of sandblasted Y-TZP demonstrated more irregular side with increasing height differences (Figure 3c,d); the average *R*_a_ was 0.70 ± 0.02 µm, average *R*_q_ was 0.85 *±* 0.04 µm, average *R*_z_ was 4.36 *±* 0.85 µm and average *R*_sk_ was −0.28 *±* 0.12 µm.

#### 3.1.3. Measurement of Contact Angle

Before the sandblasting, polished MZ had an average contact angle of 64.65° ± 0.25°, while the average contact angle was 50.48° ± 2.81° in the sandblasted surface.

### 3.2. Shear Bond Strength and Failure Mode

Table 3 shows the SBS values of experimental groups (CP0, CP1, U0, U1, CU0, CU1) and control groups (CON0, CON1). When thermocycling was not performed, there was no significant difference between the SBS of four different groups (CON0, CP0, U0, CU0; *p* > 0.05; Figure 4). After the thermocycling, the group CP1, which had sequential application of 10-MDP-containing-ceramic primer and the orthodontic primer, presented the highest SBS (*p* < 0.05; Figure 4). Thermocycling decreased the SBS except for the group CP (*p* < 0.05; Figure 4). There was no significant difference between the values of the group CON1, U1 and CU1 (*p* > 0.05).

The ARI scores are shown in Table 4. Figure 5 shows FE-SEM images of the interface between failed Y-TZP and the bracket, between failed enamel surface and the bracket. Without the thermocycling, an ARI score of 3 (Figure 5a–d) was obtained for all groups, indicating that the resin remained on the Y-TZP surface of all samples and the small amount of resin that was stuck and fell off between the retentive beads of bracket was not counted. After the thermocycling treatment, all of the samples in group CP1 still showed an ARI score of 3 (Figure 5j) but the samples in group U1 and CU1 demonstrated ARI scores of 1 (Figure 5k,l), with less than half of the resin remaining on the Y-TZP surface in 40% of the samples respectively.

## 4. Discussion

When using the 10-MDP-containing universal adhesive, two steps of application with ceramic primer and orthodontic adhesive are reduced to one step. This study evaluated the effectiveness of applying the 10-MDP-containing universal adhesive on sandblasted Y-TZP samples in order to simplify the steps of bonding between Y-TZP and the bracket. Moreover, the thermocycling treatment was performed to assess the long-term stability of bonding [24]. There are many different opinions among researchers regarding the number of thermocycling treatments. In the standard ISO 11405, it is recommended that 500 thermal cycles in water bath at temperature of 5 and 55 °C, dwell time of 20 s be made [25]. However, it has been suggested that such a recommendation may not be sufficient to reproduce the wear in an oral cavity, which led the prior researchers to increase the number of cycles [23,26,27]. According to Gale and Darvell [27], the 10,000 cycles of thermocycling corresponds to the usage period about one year in an oral cavity. The prior tests of dental bonding agent were usually performed after 5000 or 10,000 cycles of thermocycling even if the composite materials retain more than five years in an oral cavity [14,15,22]. In this study, considering the period of orthodontic treatment, the number of cycles was set to 10,000.

In this experiment, 50 μm alumina particles were sandblasted in the same manner onto every Y-TZP specimen to only evaluate the effect of the 10-MDP-containing agents. This method created micro-mechanical roughness that could increase the bonding strength between Y-TZP and the resin and it is the usual method of surface treatment in dentistry [28]. A previous study demonstrated that if alumina is sprayed on the Y-TZP surface, it generates defects on the surface and decreases the strength of Y-TZP [29]. However, Kosmac et al. [30] reported that grinding the Y-TZP surface with a diamond bur deteriorates the strength of Y-TZP, whereas sandblasting is a powerful method that strengthens Y-TZP by inducing a phase change. Moreover, Curtis et al. [31] and Karakoca and Yilmaz [32] also presented results of the bi-axial flexure strength increasing similar to those of sandblasting. In addition, Demir et al. [33] stated that sandblasting is more effective than laser irradiation for the surface treatment of zirconia. Qeblawi et al. [34] reported that if chemical conditioning is carried out after performing alumina blasting or silica blasting on the surface of Y-TZP, stable bonding can be obtained. Similarly, Kim et al. [35] also recommended the use of a 10-MDP-containing agent after alumina blasting or silica blasting. The work of Kern et al. [36] showed that when an adhesive primer was applied on the Y-TZP surface after sandblasting, the long-term bond strength was noticeably increased. Therefore, we carried out the alumina-blasting and used the functional monomer in parallel for every experimental group in our study.

Untreated Y-TZP is known to exhibit poor wettability [37]. However, if alumina particles are sandblasted on Y-TZP, the surface area and the surface energy of Y-TZP will increase, improving the wettability, thereby contributing to increased bonding-strength [38,39]. The increase in *R*_a_ from the average of 0.10 to 0.70 µm, together with increased *R*_q_ (from the average of 0.14 to 0.85 µm) and *R*_z_ (from the average of 0.96 to 4.36 µm) indicates an increase in surface area that can be available for chemical reactions. The decreased in *R*_sk_ from the average of 0.96 to −0.28 µm implies the roughness was uniformly increased when compared with polished surface with short multiple scratches. Through the surface treatment with sand blasting, the contact angle was decreased from 64.99 to 46.52 degrees, indicating an increased wettability, which allows the polymer of the resin to flow into the Y-TZP surface [38]. Based on these results, it can be said that by sandblasting the Y-TZP surface, a foundation was laid for effective resin bonding in a subsequent process.

For the control group of this study, the ceramic bracket was attached to the natural central incisor in three steps according to the routine bracket bonding protocol. The SBS was 9.59 MPa without the thermocycling and this result satisfies the bond strength of 6–8 MPa required for bonding between a tooth and a bracket [40].

There was no significant difference in SBS between groups without the thermocycling between group CP0 (9.78 ± 1.94 MPa), group U0 (9.86 ± 1.33 MPa), group CU0 (9.16 ± 0.78 MPa) and group CON0 (9.59 ± 1.77 MPa) and the bond strengths required for the bonding of bracket were satisfied. However, after the thermocycling, the SBS decreased significantly to 4.99 and 4.31 MPa in groups U1 and CU1, respectively, where the universal adhesive was applied. For groups U1 and CU1 the bonding strength generally required for the bracket bonding was not satisfied. In contrast, group CP1, with 10-MDP-containing ceramic primer applied to zirconia, retained stable SBS after thermocycling (8.16 ± 1.78 MPa). The group CP1 demonstrated the highest SBS and there was no significant difference between the other three groups (CON1, U1 and CU1). Therefore, null hypothesis could be validated.

The thermocycling treatment is an artificial aging method that is widely used in general to evaluate the long-term bonding strength. Tsuo et al. [22] reported that when an adhesive primer was applied on sandblasted Y-TZP and bonded with the resin, there was no significant difference in the SBS before and after the 10,000 cycles of thermocycling, which is consistent with the result of the present study. For group CP, with the 10- MDP-containing ceramic primer applied to the sandblasted Y-TZP, the SBS was 9.8 ± 1.8 MPa and it was slightly reduced to 8.1 ± 1.6 MPa after thermocycling, which does not represent a significant difference.

In a single-step self-etching adhesive, an acidic monomer and water are added. However, water can ionize the acidic monomer and dissolve the teeth’s calcium and phosphate ions, giving the adhesive a hydrophilic characteristic by nature. Therefore, if a single-step self-etching adhesive is placed in a moist environment, it absorbs water and because of its solubility, the moisture weakens the adhesive layer as time passes [41,42]. As the thickness of the bonding agent layer increases, the range of exposure to water increases and becomes more vulnerable to leakage. In case of the ceramic primer used in this study, the thickness of the film was so thin that it was difficult to identify the traces after application. In contrast, universal adhesive formed a rather thick film. Therefore, the results of this study can be explained by the sharper decline in SBS for Groups U1 and CU1 (with the universal adhesive applied to the surface) when compared to group CP1, where the 10-MDP-containing ceramic primer applied to the surface. However, our results also contradicted reported findings. It was reported that when sandblasted Y-TZP block was bonded with the resin after a universal adhesive (Scotchbond Universal Adhesive) was applied, high SBS was obtained even after the thermocycling treatment [43,44]. This was probably because the thermocycling was performed only 2000–2500 times, which was too low to be effective.

The ARI scores after bonding failure of sandblasted Y-TZP and the bracket are shown in Table 4 [45]. Without the thermocycling treatment, the specimens of groups CP, U and CU showed an ARI score of 3, indicating that the all adhesive left on the tooth or Y-TZP (Figure 5). This result is in line with having the stable SBS values in groups CP, U and CU (Table 3). However, after the thermocycling treatment, only group CP1 had an ARI score of 3 and groups U1 and CU1 showed that less than 50% of resin remained on the Y-TZP surface in 40% of the sample which corresponds to an ARI score of 1. This result is consistent with the decrease in SBSs of groups U1 and CU1 after the thermocycling.

To simplify the bonding steps between the Y-TZP and ceramic bracket and to obtain a stable bonding strength, this study applied the 10-MDP-containing ceramic primer or 10-MDP containing universal adhesive (or both 10-MDP-containing primer and adhesive) to the sandblasted Y-TZP and measured the SBS. The results indicated that when the 10-MDP-containing universal adhesive was used, one step was reduced in the bonding procedure and comparable SBS was obtained to bonding with enamel. However, after 10,000 rounds of thermocycling, the bonding strength decreased significantly when the 10-MDP-containing universal adhesive was used. Therefore, considering that an orthodontic treatment is continued for a long period after the bracket bonding, it is preferable to apply the 10-MDP-containing ceramic primer to the sandblasted Y-TZP, followed by application of the orthodontic primer and bonding of orthodontic bracket using the orthodontic composite resin.

## 5. Conclusions

In the present study, the long term stability of shear bond strength was evaluated when 10-MDP-containing universal adhesive was used in the ceramic bracket bonding on dental zirconia.

In ceramic bracket bonding, the use of a universal adhesive showed a clinically acceptable range of the shear bond strength after 24 h. After artificial aging process, the shear bond strength of the universal adhesive was more affected when compared with using a ceramic primer and an orthodontic primer.

For clinical aspects, it can be concluded that, when bonding ceramic brackets to a dental zirconia, the application of the ceramic primer on the zirconia followed by an orthodontic primer is recommended rather than using a universal adhesive.

## Figures and Tables

**Figure 1 materials-12-01640-f001:**
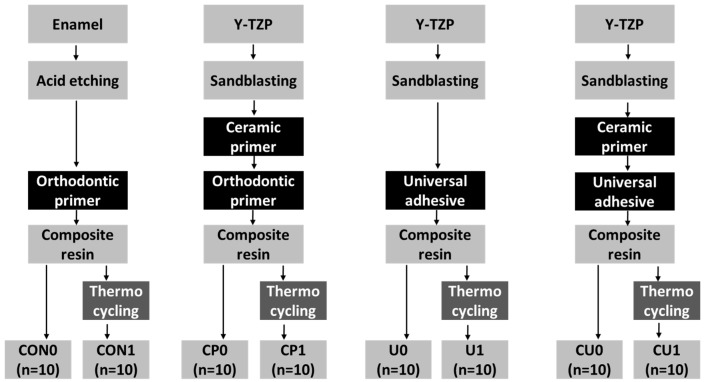
Flow chart of the experiment.

**Figure 2 materials-12-01640-f002:**
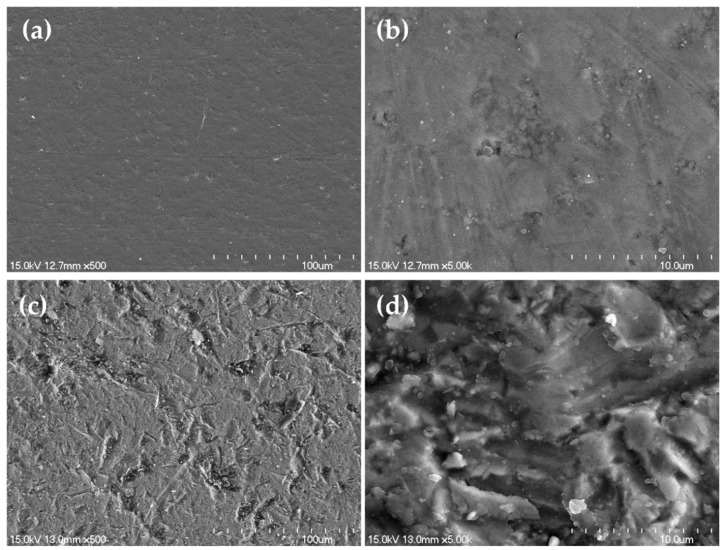
Field emission scanning electron microscope (FE-SEM) images of surface of yttrium-stabilized tetragonal zirconia polycrystal (Y-TZP) before; ((**a**) ×500; (**b**) ×5000) and after alumina sandblasting; ((**c**) ×500; (**d**) ×5000).

**Figure 3 materials-12-01640-f003:**
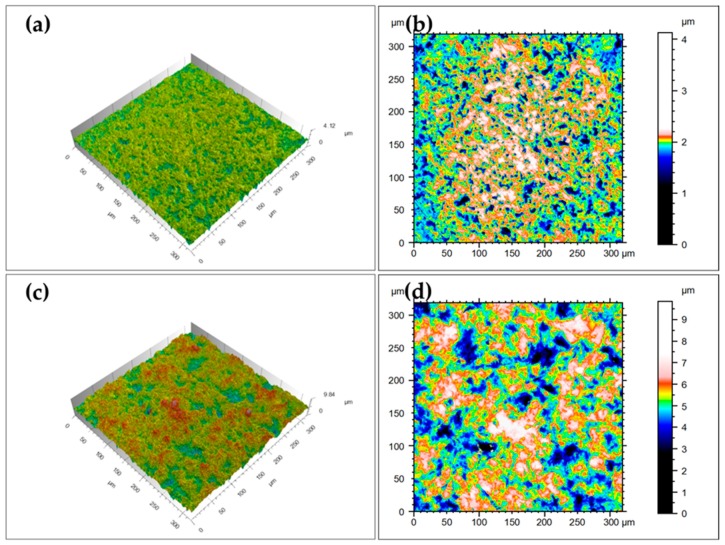
Confocal laser scanning microscope (CLSM) images of zirconia surface before (**a**,**b**) and after (**c**,**d**) the alumina sandblasting. The height difference is graded in colour (**b**,**d**).

**Figure 4 materials-12-01640-f004:**
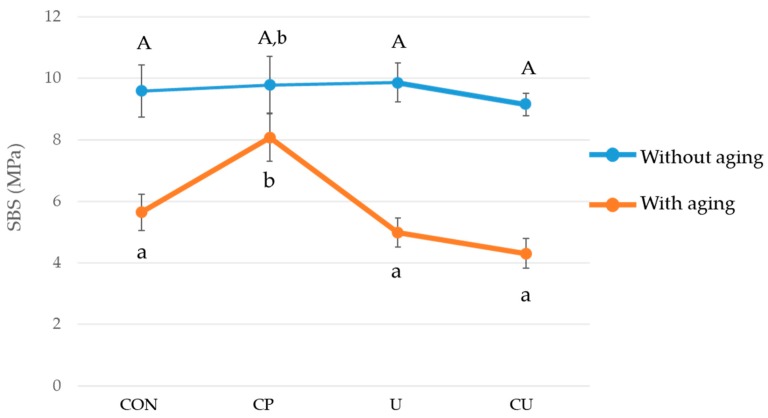
Shear bond strength values obtained when orthodontic ceramic bracket was bonded to enamel (CON0, CON1) or Y-TZP (CP0, CP1, U0, U1, CU0, CU1), with or without thermocycling (aging). The vertical bars indicate the standard deviations. There are significant differences in SBS between the groups marked with different letters A, a and b (*p* < 0.05).

**Figure 5 materials-12-01640-f005:**
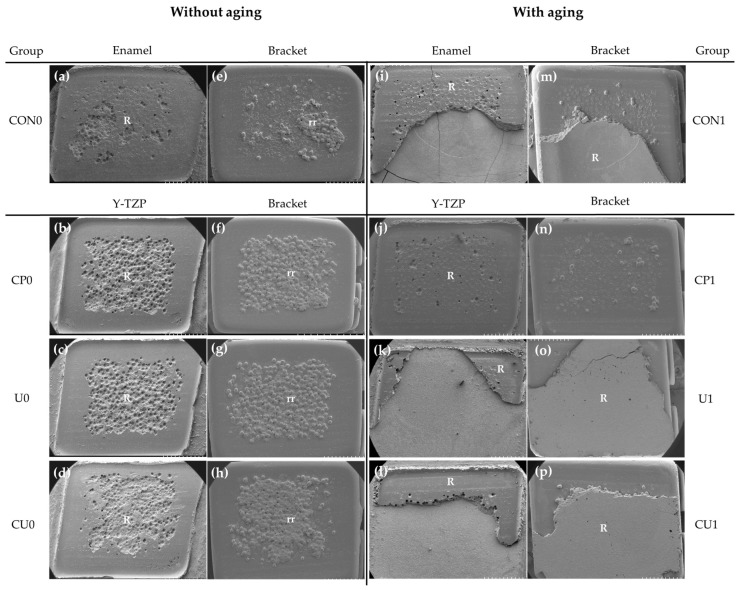
Images of failure interface (×30) without- and with aging: (**a**,**i**) enamel surface; (**b**–**d**,**j**–**l**) Y-TZP surface; (**e**–**h**) and (**m**–**p**) debonded bracket surface. Abbreviations: R, resin; rr, resin captured by retentive beads of bracket.

**Table 1 materials-12-01640-t001:** Materials used in this study.

Material	Product Name	LOT Number	Main Component	Manufacturer
Zirconia	LAVA Plus	515920	Tetragonal polycrystalline zirconia, 3 mol% yttria, alumina	3M ESPE, USA
Primer	Clearfil ceramic primer	240010	3-Methacryloxypropyl trimethoxy silane, ^a^ 10-MDP, ethanol	Kuraray, Japan
Adhesive	Transbond XT adhesive primer	ER7BS	^b^ TEGDMA, ^c^ bis-GMA, triphenylantimony, 4-(dimethylamino)-benzeneethanol, dl-camphorquinone, hydroquinone	3M Unitek, USA
	Clearfil S^3^ Bond	170008	10-MDP ^a^, ^c^ bis-GMA, ^d^ HEMA, hydrophobic demethacrylate, dl-camphorquinone, ethyl alcohol, water, silanated colloidal silica	Kuraray, Japan
Composite Resin	Transbond XT adhesive paste	ER7BS	Silane treated quartz, ^c^ bis-GMA, bisphenol A bis (2-hydroxyethyl ether) dimethacrylate, silane-treated silica	3M Unitek, USA
Etchant	Scotchbond Universal Etchant	577060	Water, phosphoric acid, synthetic amorphous silica, fumed, polyethylene glycol, aluminium oxide	3M ESPE, USA

Abbreviations: ^a^ 10-MDP, 10-methacryloyloxydecyl dihydrogen phosphate; ^b^ TEGDMA, triethylene glycol dimethacrylate; ^c^ bisGMA, bisphenol-A-diglycidylether dimethacrylate; ^d^ HEMA, hydroxyethyl methacrylate.

**Table 2 materials-12-01640-t002:** Adhesive remnant index (ARI) score and criterion.

ARI Score	Criterion
0	No adhesive left on the tooth
1	Less than half of the adhesive left on the tooth
2	More than half of the adhesive left on the tooth
3	All adhesive left on the tooth, with a distinct impression of the bracket mesh
4	Enamel fracture

**Table 3 materials-12-01640-t003:** Comparison of shear bond strength (mean ± SD (range)) of all groups.

Group	Shear Bond Strength (MPa)	Group	Shear Bond Strength (MPa)
CON0	9.59 ± 1.77 (8.32–10.86)	CON1	5.65 ± 1.24 (4.76–6.54)
CP0	9.78 ± 1.94 (8.40–11.18)	CP1	8.16 ± 1.78 (6.88–9.43)
U0	9.86 ± 1.33 (8.90–10.81)	U1	4.99 ± 0.99 (4.28–5.70)
CU0	9.16 ± 0.78 (8.60–9.71)	CU1	4.31 ± 1.02 (3.58–5.03)

**Table 4 materials-12-01640-t004:** The percentage (%) of ARI scores in the groups after shear bond strength test.

Without Aging	ARI Score				With Aging	ARI Score			
	0	1	2	3		0	1	2	3
CON0	–	20	–	80	CON1	10	20	10	60
CP0	–	–	–	100	CP1	–	–	–	100
U0	–	–	–	100	U1	–	40	–	60
CU0	–	–	–	100	CU1	–	40	–	60

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
