# Peer review of "Effect of Simplified Bonding on Shear Bond Strength between Ceramic Brackets and Dental Zirconia"

_materials, 2019, doi:10.3390/ma12101640_

Round 1
Reviewer 1 Report
You must take into account the suggestions and changes requested.
The article should be reviewed by a native English-speaking specialist.
The conclusions should be rewritten.

Author Response
Point 1. You must take into account the suggestions and changes requested.
Response 1: Thank you for your comment. The manuscript was revised by following your recommendation.
Point 2. The article should be reviewed by a native English-speaking specialist.
Response 2: Thank you for your comment. The manuscript was reviewed by an English-speaking specialist.
Point 3. The conclusions should be rewritten.
Response 3: Thank you for your comment. The conclusion was rewritten.
Before: In this study, we applied 10-MDP-containing primer and adhesive to sandblasted Y-TZP to evaluate the effect of 10-MDP adhesive on the bonding of Y-TZP and ceramic brackets to simplify the bonding step. The following conclusions were drawn.
When the 10-MDP-containing adhesive was used only, acceptable SBS was obtained while simplifying the bonding step.
The long-term stable SBS values was obtained when the 10-MDP-containing primer followed by Transbond XT adhesive primer was applied to Y-TZP.
After: Within the limitations of this study, the null hypothesis was accepted. Based on the results, when bonding ceramic brackets to dental zirconia surface, we can conclude that ceramic primer used with an orthodontic primer, rather than using an universal adhesive is recommended.
Point 4. Improve English translation as well as abstract writing. The abstract is very confusing.
Response 4: Thank you for your comment. The abstract was rewritten and the manuscript was revised for improvement of english translation. The name of groups was changed as follows:
Before | After |
CON | CON0 |
CONT | CON1 |
C | CP0 |
CT | CP1 |
S | U0 |
ST | U1 |
CS | CU0 |
CST | CU1 |
Before: With the increase in adult orthodontic patients, it is common to bonding a ceramic bracket to the anterior zirconia restoration. In this study, we investigated whether the adhesion between a ceramic bracket and yttrium-stabilized tetragonal zirconia polycrystal (Y-TZP) can be simplified by using a 10-MDP-containing adhesive. In the control group, bracket bonded on maxillary central incisor. All Y-TZPs were polished and sandblasted; 10-MDP-containing primer and orthodontic primer were applied in group C, 10-MDP-containing adhesive was applied in group S, and both 10-MDP-containing primer and adhesive were applied in group CS. A ceramic bracket was bonded to every Y-TZP with orthodontic resin. SBS was measured after with or without thermocycling. The data were statistically analyzed by using the Kruskal–Wallis test. Without thermocycling (C, S, CS), the SBS was within the acceptable range and showed adhesive failure between resin and bracket in all groups. With aging (CT, ST, CST), group CT showed the highest and reliable SBS and exhibited adhesive failure between resin and bracket. This demonstrates that with a simplified bonding step (S), stable bond strength of Y-TZP and resin can be obtained, however, the long-term stable SBS were obtained in group CT.
After: The aim of this study was to evaluate the long term stability of shear bond strength (SBS) when 10-MDP-containing universal adhesive was used in the ceramic bracket bonding on dental zirconia. Twenty human maxillary incisors were collected. The ceramic bracket was bonded on the buccal enamel surface after the acid-etching and orthodontic primer application (Group CON). Sixty zirconia specimens were sintered, sandblasted, and divided into three experimental groups; group CP, ceramic primer followed by an orthodontic primer; group U, universal adhesive; group CU, ceramic primer followed by an universal adhesive. For each specimen, the bracket was bonded on the treated surface with composite resin (Transbond XT, 3M ESPE). The shear bond strength (SBS) test was tested before (CON1, CP1, U1, CU1) and after the artificial aging (CON0, CP0, U0, CU0). The data were statistically analyzed with the Kruskal–Wallis test at a significance level of 0.05. The mean SBS of CON0, CP0, U0, and CU0 were within the clinically acceptable range without significant differences. After the aging process, SBS were decreased in all groups. Among the aged groups, CP1 showed the highest SBS. Based on the results, when bonding ceramic brackets to dental zirconia surface, we can conclude that ceramic primer used with an orthodontic primer, rather than using universal adhesive is recommended.
Point 5. 74: null hypotheses
Response 5: Thank you for your comment. The part of manuscript was revised.
Before: The hypotheses tested in this study are as follows: (1) Applying 10-MDP-containing adhesive to the sandblasted Y-TZP surface does not maintain the stable SBS of resin and Y-TZP; (2) Applying 10-MDP-containing primer to the sandblasted Y-TZP surface does not maintain the stable SBS of resin and Y-TZP
After: The null hypotheses tested were as follows: Applying 10-MDP-containing universal adhesive to the sandblasted zirconia surface did not increased SBS between the ceramic bracket and zirconia.
Point 6. 92: How many? What type? What were the reasons for the extractions?
Response 6: Thank you for your comment. Twenty maxillary central incisors extracted due to hopeless statement, i.e., severe periodontal disease were used in this experiment. The part of manuscript was revised.
Before: Extracted human teeth were immersed in a 0.1% thymol solution for a week and stored at 4 °C.
After: Sound human maxillary central incisors (n=20) extracted for periodontal reasons were used under a protocol approved by an Institutional Review Board (No. S-D20150041).
Point 7. 93: Do you have any document from the ethics committee?
Response 7: Yes, it’ll be attached to you with the reply.
Point 8. Do you have any patients informed consent?
Response 8: In our opinions, it is not necessary because this study does not include any information related to the patient.
Point 9. 133: Why did you only use the Ra? Why you didn’t selected the Rz, Rq and Rsk. Ra and Rq arithmetic average and root mean square, respectively? Due to their large use and also permit to compare sample roughness in what concerns their esthetic aspects. Rz average distance between the highest peak and lowest valley in each sampling length enables to have an idea of the profile total high. The skewness parameter, Rsk, is indicated to relate the surface load capability.
Please consult the article: M. Carreira, P. V. Antunes, A. Ramalho, A. Paula, E. Carrilho.Thermocycling effect on mechanical and tribological characterization of two indirect dental restorative materials J Braz. Soc. Mech. Sci. Eng. DOI 10.1007/s40430-016-0579-6 DO NOT FEEL PRESSED TO CIT THIS ARTICLE
Response 9: Thank you for your comment and references. As your advice, the roughness parameters have been added to the manuscript.
Before: The roughness (Ra) of the surface before and after sandblasting, were determined by using a 3D surface confocal laser scanning microscope (CLSM; LSM 800-MAT, Carl Zeiss MicroImaging GmbH, Germany).
After: The roughness (Ra, Rq, Rz, Rsk) of the surface before and after sandblasting were determined by using a 3D surface confocal laser scanning microscope (CLSM; LSM 800-MAT, Carl Zeiss MicroImaging GmbH, Germany). The roughness parameters mean: Ra, average surface roughness; Rq, root mean square roughness; Rz, mean roughness depth; and Rsk, roughness skewness.
Point 10. 151: "kruskal wallis anova" does not exist. Kruskal-Wallis is an alternative method to One-way ANOVA.
Response 10: Thank you for your comment. The part of manuscript was revised.
Before: Statistical analysis was performed using Kruskal–Wallis ANOVA..
After: Shear bond strength was compared between experimental groups with Kruskal–Wallis test.
Point 11. 221: You didn't followed the ISO 11405 standard. For a long time the references indicate the lack of effect of these times and number of cycles on the mechanical and tribological properties of the substrates and of varied materials. Since 1999 Gale and Darvel; 2014 Moresi et al; 2016 Carrera and Carrilho they don't adopt the use of the ISO norm. It would also be desirable to refer more and recent publications
Response 11: Thank you for your comment and references. The part of manuscript was revised.
Before: There are many different opinions among researchers regarding the number of thermocycling treatments, but considering that an orthodontic treatment is usually carried out over a period of one year, the number of cycles was set to 10,000 in this experiment, which corresponds to the usage period of about one year in an oral cavity
After: In the standard ISO 11405, it is recommended that 500 thermal cycles in water bath at temperature of 5 and 55 °C, dwell time of 20 s be made [25]. However, it has been suggested that such a recommendation may not sufficient to reproduce the wear in an oral cavity, which led the prior researchers to increase the number of cycles [23, 26, 27]. According to Gale and Darvell [27], the 10,000 cycles of thermocycling corresponds to the usage period about one year in an oral cavity. The prior tests of dental bonding agent were usually performed after 5,000 or 10,000 cycles of thermocycling even if the composite materials retain more than five years in an oral cavity [14, 15, 22]. In this study, considering the period of orthodontic treatment, the number of cycles was set to 10,000.
Point 12. 261: Discuss if they used or not the ISO norm. If not, they used the same number of cycles?
Response 12: Yes, they used the same number of cycles (10,000). The part of manuscript was revised.
Before: Tsuo et al. reported that when an adhesive primer was applied on sandblasted Y-TZP and bonded with the resin, there was no significant difference in the SBS before and after the thermocycling treatment
After: Tsuo et al. [22] reported that when an adhesive primer was applied on sandblasted Y-TZP and bonded with the resin, there was no significant difference in the SBS before and after the 10,000 cycles of thermocycling, which is consistent with the result of the present study.
Point 13. 304: There is no need to have this sentence present in the conclusions. You should only confirm or not the null hypotheses
Response 13: Thank you for your comment. The part of manuscript was revised.
Before: In this study, we applied 10-MDP-containing primer and adhesive to sandblasted Y-TZP to evaluate the effect of 10-MDP adhesive on the bonding of Y-TZP and ceramic brackets to simplify the bonding step. The following conclusions were drawn.
When the 10-MDP-containing adhesive was used only, acceptable SBS was obtained while simplifying the bonding step.
The long-term stable SBS values was obtained when the 10-MDP-containing primer followed by Transbond XT adhesive primer was applied to Y-TZP.
After: Within the limitations of this study, the null hypothesis was accepted. Based on the results, when bonding ceramic brackets to dental zirconia surface, we can conclude that ceramic primer used with an orthodontic primer, rather than using an universal adhesive is recommended.
Point 14. 311: "...it will be necessary to conduct an 300 additional study to focus on obtaining stable long-term bonding strength while reducing the number 301 of bonding steps..." replace this conclusion in the conclusion space.
Response 14: Thank you for your comment. The part of the manuscript was excluded.
Before: Also, it will be necessary to conduct an additional study to focus on obtaining stable long-term bonding strength while reducing the number of bonding steps.
After: The part of the manuscript was excluded.
Reviewer 2 Report
An extensive English correction thought the document must be made – I recommend the authors to seek an English native or an English translator to review the manuscript
Please check the title for English error – an article must be introduced before ceramic or brackets must be put in the plural form
Please substitute orthodontic patients for patients performing/under orthodontic treatment
The abstract must be rewritten, the text is confusing, and some phrases are out of context
Abbreviations can only be used after the word has been written unabbreviated the first time. Please correct situations like this, for example in line 22 SBS appears abbreviated but no unabbreviated word is seen before.
The introduction needs to be rewritten. A logical line must be followed through the text. Several phrases are wrongly positioned at the text, and the reader gets lost.
Line 33: the number of patients performing an orthodontic treatment is increasing for several reasons, not only by the attention paid to aesthetics. Please rewrite and include other reasons.
Line 34-36: the two sentences “The use of bracket…to the anteriors for aesthetics” refer to the same thing. Please merge them in only one statement
Line 39: "in the past, patients usually choose conventional porcelain fused to the metal crown…" In my opinion, it is incorrect to refer to this was only a patient choice. Other factors as laboratory work, ceramics with weak physical properties… also, contribute to the choice for PFM works. Please rewrite.
Since the authors refer at lines 66-68 that 10-MDP is known to improve the bonding strength between Y-TZP and resin, why the hypotheses tested are written in the negative form? That applying 10-MDP-containing adhesive does not maintain the stable SBS of resin and Y-TZP?
Line 80: please change to zirconia
Line 81: please insert the specimens’ dimensions and shape
Lines 91-92: if the extracted tooth were immersed in a 0,1% thymol solution for a week, the teeth extracted for more than one week and less than 3 months, as stated, were kept in what solution? Please clarify.
Lines 96-98: the authors refer that teeth were cleaned with distilled water for 2 min and then dried, followed by treatment with 37% phosphoric acid. Any precaution was performed to avoid excessive drying of the tooth? Since we know excessive drying alters tooth structure and adhesion.
Line 117: why the authors choose a total curing period of 40s? it is superior to manufacturer instructions. Please justify.
Table 1: please add the n tested for each group
Lines 128-130: more detailed information should be added regarding how observation of microstructure of brackets was performed (what characteristics were evaluated)
Lines 151-152: The Kruskal-Wallis and the ANOVA tests are two different tests. Please reformulate this description and add more information about who the tests were chosen
To facilitate readers comprehension, a logical correspondence should be made between the materials and methods and the results section. For example, section 2.4 of methods – surface characteristics of Y-TZP should have a correspondent section in results. The authors divided this section in two, 3.2 – Surface roughness of Y-TZP and 3.3 – Contact angle on Y-TZP, which can be confusing for the readers. Please reformulate.
Why evaluation of bracket microstructure is relevant? Why did authors choose to evaluate parameters like strength, color or height? Add that information in the manuscript, why it is important for the aim of the work and discuss the obtained results.
Lines 164-168: please add figure 2a at the end of the phrase: “…was found” and figure 2b at the end of the phrase “…were observed” to facilitate readers comprehension. Repeat for figures 2c and 2d.
Line 171: Replace ; for : after alumina
Line 180: Replace ; for : after alumina
The authors state in lines 219-222 that orthodontic treatment is performed generally for a period of one year. This is not correct, orthodontic treatments are usually performed for a period of 18 to 24 months. Please rewrite this statement and justify the 10,000 cycles used in the work
Lines 245-246: The authors must discuss why an increase in wettability is better for adhesion
The discussion should be improved, focusing more on the experiments performed and the obtained results. More detailed integration of results should be made.
References must be revised, and more actual references should be added to the manuscript
Author Response
Point 1. An extensive English correction thought the document must be made – I recommend the authors to seek an English native or an English translator to review the manuscript
Response 1: Thank you for your comment. The manuscript was revised by English translator.
Point 2. Please check the title for English error – an article must be introduced before ceramic or brackets must be put in the plural form
Response 2: Thank you for your comment. The part of the manuscript was revised by following your recommendation.
Before: Effect of Simplified Bonding on Shear bond strength between ceramic bracket and dental zirconia.
After: Effect of Simplified Bonding on Shear bond strength between ceramic brackets and dental zirconia.
Point 3. Please substitute orthodontic patients for patients performing/under orthodontic treatment
Response 3: Thank you for your comment. The part of the manuscript was revised by following your recommendation.
Before: The number of adult orthodontic patients is increasing as rising attention is paid to aesthetics.
After: The number of adult patients performing an orthodontic treatment is increasing for several reasons, not only by the attention paid to aesthetics, but also improvement of bite.
Point 4. The abstract must be rewritten, the text is confusing, and some phrases are out of context
Response 4: Thank you for your comment. The abstract has been revised.
Before: With the increase in adult orthodontic patients, it is common to bonding a ceramic bracket to the anterior zirconia restoration. In this study, we investigated whether the adhesion between a ceramic bracket and yttrium-stabilized tetragonal zirconia polycrystal (Y-TZP) can be simplified by using a 10-MDP-containing adhesive. In the control group, bracket bonded on maxillary central incisor. All Y-TZPs were polished and sandblasted; 10-MDP-containing primer and orthodontic primer were applied in group C, 10-MDP-containing adhesive was applied in group S, and both 10-MDP-containing primer and adhesive were applied in group CS. A ceramic bracket was bonded to every Y-TZP with orthodontic resin. SBS was measured after with or without thermocycling. The data were statistically analyzed by using the Kruskal–Wallis test. Without thermocycling (C, S, CS), the SBS was within the acceptable range and showed adhesive failure between resin and bracket in all groups. With aging (CT, ST, CST), group CT showed the highest and reliable SBS and exhibited adhesive failure between resin and bracket. This demonstrates that with a simplified bonding step (S), stable bond strength of Y-TZP and resin can be obtained, however, the long-term stable SBS were obtained in group CT.
After: The aim of this study was to evaluate the long term stability of shear bond strength (SBS) when 10-MDP-containing universal adhesive was used in the ceramic bracket bonding on dental zirconia. Twenty human maxillary incisors were collected. The ceramic bracket was bonded on the buccal enamel surface after the acid-etching and orthodontic primer application (Group CON). Sixty zirconia specimens were sintered, sandblasted, and divided into three experimental groups; group CP, ceramic primer followed by an orthodontic primer; group U, universal adhesive; group CU, ceramic primer followed by an universal adhesive. For each specimen, the bracket was bonded on the treated surface with composite resin (Transbond XT, 3M ESPE). The shear bond strength (SBS) test was tested before (CON1, CP1, U1, CU1) and after the artificial aging (CON0, CP0, U0, CU0). The data were statistically analyzed with the Kruskal–Wallis test at a significance level of 0.05. The mean SBS of CON0, CP0, U0, and CU0 were within the clinically acceptable range without significant differences. After the aging process, SBS were decreased in all groups. Among the aged groups, CP1 showed the highest SBS. Based on the results, when bonding ceramic brackets to dental zirconia surface, we can conclude that ceramic primer used with an orthodontic primer, rather than using an universal adhesive is recommended.
Point 5. Abbreviations can only be used after the word has been written unabbreviated the first time. Please correct situations like this, for example in line 22 SBS appears abbreviated but no unabbreviated word is seen before.
Response 5: Thank you for your comment. The abbreviations have been revised.
Before: SBS was measured after with or without thermocycling.
After: The shear bond strength (SBS) test was tested before (CON1, CP1, U1, CU1) and after the artificial aging (CON0, CP0, U0, CU0).
Point 6. The introduction needs to be rewritten. A logical line must be followed through the text. Several phrases are wrongly positioned at the text, and the reader gets lost.
Response 6: Thank you for your comment. The introduction of the manuscript was revised extensively by following your recommendation. Please refer to the attached file for revised introduction.
Point 7. Line 33: the number of patients performing an orthodontic treatment is increasing for several reasons, not only by the attention paid to aesthetics. Please rewrite and include other reasons.
Response 7: Thank you for the comment. Most of the reason was about aesthetics such as improvement of smile line or facial appearance. The part of manuscript was revised by following your recommendation
Before: The number of adult orthodontic patients is increasing as rising attention is paid to aesthetics
After: The number of adult patients performing an orthodontic treatment is increasing for several reasons, not only by the attention paid to aesthetics, but also demand for the improvement of the occlusal relationships [12, 13]
Point 8. Line 34-36: the two sentences “The use of bracket…to the anteriors for aesthetics” refer to the same thing. Please merge them in only one statement
Response 8: Thank you for your comment. The part of the manuscript was revised.
Before: The use of bracket materials like ceramics instead of conventional metals is also increasing for the aesthetic appearance attached to teeth [2]. A ceramic bracket is usually attached to the anteriors for aesthetics
After: The use of ceramic brackets is also increasing for the improved aesthetics [13].
Point 9. Line 39: "in the past, patients usually choose conventional porcelain fused to the metal crown…" In my opinion, it is incorrect to refer to this was only a patient choice. Other factors as laboratory work, ceramics with weak physical properties… also, contribute to the choice for PFM works. Please rewrite.
Response 9: Thank you for your comment. The part of the manuscript was removed and substituted.
Before: In the past, patients usually chose conventional porcelain fused to the metal crown (PFM) as the anterior restoration material, but because of issues such as fracture of porcelain or darkening of gingival shade owing to the metal lower structure, the use of MZ is increasing
After: The dental zirconia offers excellent mechanical strength, and provides optimal esthetics compared to porcelain fused to metal restorations [1, 2].
Point 10. Since the authors refer at lines 66-68 that 10-MDP is known to improve the bonding strength between Y-TZP and resin, why the hypotheses tested are written in the negative form? That applying 10-MDP-containing adhesive does not maintain the stable SBS of resin and Y-TZP?
Response 10: Thank you for your comment. The negative null hypothesis was used since the long term stability of hydrophilic universal adhesive was unclear when used in bracket bonding.
Before: (1) Applying 10-MDP-containing adhesive to the sandblasted Y-TZP surface does not maintain the stable SBS of resin and Y-TZP; (2) Applying 10-MDP-containing primer to the sandblasted Y-TZP surface does not maintain the stable SBS of resin and Y-TZP.
After: The null hypotheses tested were as follows: Applying 10-MDP-containing universal adhesive to the sandblasted zirconia surface did not increased SBS between the ceramic bracket and zirconia
Point 11. Line 80: please change to zirconia
Response 11: Thank you for your comment. The part of the manuscript was revised.
Before: Zircnoia
After: Zirconia
Point 12. Line 81: please insert the specimens’ dimensions and shape
Response 12: Thank you for your comment. The information was added to the manuscript.
Before: A total of 60 rectangular monolithic zirconia specimens were prepared from a green-stage Y-TZP block (LAVA Plus, 3M ESPE, USA) that was sintered according to the manufacturer’s instructions.
After: Sixty cuboidal Y-TZP specimens in 15 (width) × 15 (height) × 3 (thickness) mm were prepared from a green-stage Y-TZP block (LAVA Plus, 3M ESPE, St. Paul, MN, USA) and sintered according to the manufacturer’s instructions.
Point 13. Lines 91-92: if the extracted tooth were immersed in a 0,1% thymol solution for a week, the teeth extracted for more than one week and less than 3 months, as stated, were kept in what solution? Please clarify.
Response 13: Thank you for your comment. It was a mistaken sentence. The part of the manuscript was revised.
Before: Extracted human teeth were immersed in a 0.1% thymol solution for a week and stored at 4 °C. Prior to the experiments, the teeth had been stored for less than 3 months, and they were used for the study with the approval of the School of Dentistry, Seoul National University Institutional Review Board (No. S-D20150041).
After: Sound human maxillary central incisors (n=20) extracted for periodontal reasons were used in this experiment under a protocol approved by an Institutional Review Board (No. S-D20150041). Maxillary central incisors (n = 20) which had been stored in a 0.1% thymol solution at 4 °C for less than 3 months after extraction, were used.
Point 14. Lines 96-98: the authors refer that teeth were cleaned with distilled water for 2 min and then dried, followed by treatment with 37% phosphoric acid. Any precaution was performed to avoid excessive drying of the tooth? Since we know excessive drying alters tooth structure and adhesion.
Response 14: Thank you for your comment. Bonding on the “enamel” surface, abundant drying process is inevitable when compared with dentin surface. When the dentin surface is completely dried, the bonding failure occurs because of the internal structure like collagen network collapse after the drying process, but it rarely happens in enamel. The part of the manuscript was revised.
Before: It was cleaned with distilled water for 2 min by using an ultrasonic machine and then dried, and then the labial surfaces were treated by using 37% phosphoric acid etchant (Scotchbond Universal Etchant, 3M ESPE, USA) according to the manufacturers’ instructions.
After: The excessive moisture was dried with air by using three-way syringe. The enamel was etched with 37% phosphoric acid (Scotchbond Universal Etchant, 3M ESPE, USA) for 20 s and then washed for 20 s using water, followed by drying with gentle air from the three-way syringe according to the manufacturers’ instructions.
Point 15. Line 117: why the authors choose a total curing period of 40s? it is superior to manufacturer instructions. Please justify.
Response 15: Thank you for your comment. Unlike light curing in direct resin restorations, the bulky structure of the bracket inhibits the curing efficacy so the prolonged curing time was adopted. The part of manuscript was revised by following your recommendation.
Before: Excessive resin was removed with a resin applicator, and the bracket was cured with a light-emitting diode (LED) curing unit (Elipar Free Light 2, 3M ESPE, USA) at each margin for 10 s (total curing period: 40 s).
After: Light-curing was performed toward the margin of the bracket with a light-emitting diode (LED) curing unit (Elipar Free Light 2, 3M ESPE, USA). Light-cured at each margin for 10 s, therefore light-curing was performed for a total of 40 s. Since the composite resin is located between the bracket and the Y-TZP, a sufficient amount of light was irradiated instead of following the manufacturer’s recommendation (10 s) to prevent any error due to insufficient polymerization of the composite resin.
Point 16. Table 1: please add the n tested for each group
Response 16: Thank you for your comment. The Table1 was substituted for Figure1.
Before: Table 1. Experimental plan of this study.
Step 1. Pre-treatment | E | S | S | S | E | S | S | S |
Step 2. Primer | – | CP | – | CP | – | CP | – | CP |
Step 3. Adhesive | XTp | XTp | SB | SB | XTp | XTp | SB | SB |
Bracket bonding | XTa | XTa | XTa | XTa | XTa | XTa | XTa | XTa |
Thermocycling | N | N | N | N | Y | Y | Y | Y |
Group | CON | C | S | CS | CONT | CT | ST | CST |
After: Please refer to the attached file (figure1.)
Point 17. Lines 128-130: more detailed information should be added regarding how observation of microstructure of brackets was performed (what characteristics were evaluated)
Response 17: Thank you for your comment. The part of manuscript was revised. In the revised manuscript, observation of bracket surface was excluded because it was considered to be less relevant to the present study.
Before: 2.3. Observation of Microstructure of bracket
The topography of the bracket was observed by using topography analyzer (OLS 5000, Olympus, Japan).
After: This part was removed.
Point 18. Lines 151-152: The Kruskal-Wallis and the ANOVA tests are two different tests. Please reformulate this description and add more information about who the tests were chosen
Response 18: Thank you for your comment. The part of manuscript was revised.
Before: Statistical analysis was performed using Kruskal–Wallis ANOVA and the Statistical Package for the Social Sciences 22.0 (SPSS, IBM, USA).
After: Shear bond strength was compared between experimental groups with Kruskal–Wallis test. The Statistical Package for the Social Sciences (version 22.0; IBM, USA) was used, with a significance level of 5%.
Point 19. To facilitate readers comprehension, a logical correspondence should be made between the materials and methods and the results section. For example, section 2.4 of methods – surface characteristics of Y-TZP should have a correspondent section in results. The authors divided this section in two, 3.2 – Surface roughness of Y-TZP and 3.3 – Contact angle on Y-TZP, which can be confusing for the readers. Please reformulate.
Response 19: Thank you for your comment. The section of the manuscript was revised.
Before:
2. Materials and methods
2.1. Specimen Preparation
2.1.1. Zircnoia
2.1.2. Tooth
2.2. Surface treatment and Bracket Bonding
2.3. Observation of Microstructure of bracket
2.4. Surface charateristics of Y-TZP
2.5. Shear Bond Testing and Observation of Failure Mode
3. Result
3.1. Microstructure of bracket
3.2. Surface Roughness of Y-TZP
3.3. Contact Angle on Y-TZP
3.4. Shear Bond Strength and Failure Mode
After:
2. Materials and Methods
2.1. Preparation of Y-TZP
2.2. Observation of surface characteristics of Y-TZP
2.2.1. Field emission-scanning electron microscopic examination
2.2.2. Measurement of surface roughness
2.2.3. Measurement of contact angle
2.3. Bonding of bracket and shear bond strength test
2.3.1. Experimental design and Bonding of the bracket to Y-TZP
2.3.2. Shear bond strength test
3. Results
3.1. Surface Characteristics of Y-TZP
3.1.1. Field emission-scanning electron microscopic examination
3.1.2. Measurement of surface roughness
3.1.3. Measurement of contact angle
3.2. Shear bond strength and failure mode
Point 20. Why evaluation of bracket microstructure is relevant? Why did authors choose to evaluate parameters like strength, color or height? Add that information in the manuscript, why it is important for the aim of the work and discuss the obtained results.
Response 20: Thank you for your comment. In the revised manuscript, observation of bracket surface was excluded because it was considered to be less relevant to the present study.
Before: The strength, color, height, and map of the bracket were examined by analyzing the topography of bracket, as shown in Figure 1. The strength was higher at the part with attached beads than at the border of the bracket, and the height increased as the distance from the center to the outer edges on both sides increased.
After: à This part of the the result was removed.
Point 21. Lines 164-168: please add figure 2a at the end of the phrase: “…was found” and figure 2b at the end of the phrase “…were observed” to facilitate readers comprehension. Repeat for figures 2c and 2d.
Response 21: Thank you for your comment. The part of the manuscript was revised.
Before: When the polished Y-TZP sample was examined at ×500 magnification, a very even surface was found; when it was examined at ×5,000 magnification, some directionless scratches were observed (Figure 2a,b). Examination of the sandblasted Y-TZP sample at ×500 magnification revealed uneven and roughly dented traces scattered on the surface. Meanwhile, the ×5,000 image shows a much rougher surface than that of the polished sample (Figure 2c,d).
After: When the polished, pre-sandblasted Y-TZP specimen was examined at ×500, a very even surface was found (Figure 2a); when it was examined at ×5,000, some directionless scratches were observed (Figure 2b). Examination of the sandblasted Y-TZP specimen at ×500 revealed uneven and roughly dented traces scattered on the surface (Figure 2c). Meanwhile, the ×5,000 image shows much rougher surface than that of the polished sample (Figure 2d).
Point 22. Line 171: Replace ; for : after alumina
Response 22: Thank you for your comment. The part of the manuscript was revised.
Before: Figure 2. SEM images of Y-TZP surfaces after polishing with 500 grit diamond wheel: (a) ×500 magnification; (b) ×5000 magnification and after sandblasting with 50 μm alumina; (c) ×500 magnification; (d) ×5000 magnification
After: Figure 2. FE-SEM images of surface of Y-TZP before sandblasting: (a, ×500; b, ×5000) and after alumina sandblasting; (c, ×500; d, ×5000).
Point 23. Line 180: Replace ; for : after alumina
Response 23: Thank you for your comment. The part of the manuscript was revised.
Before: Figure 3. CLSM images of Y-TZP surfaces after grinding with 500 grit diamond disc: (a) three-dimensional images; (b) flat images with height difference in color and after air-abrasion with 50 μm alumina; (c) three-dimensional images; (d) flat images with height difference in color.
After: Figure 3. CLSM images of polished, pre-sandblasted Y-TZP (a, 3D image and b, flat image with height difference in color); CLSM images of sandblasted Y-TZP (c, 3D image and d, flat image with height difference in color.
Point 24. The authors state in lines 219-222 that orthodontic treatment is performed generally for a period of one year. This is not correct, orthodontic treatments are usually performed for a period of 18 to 24 months. Please rewrite this statement and justify the 10,000 cycles used in the work
Response 24: Thank you for your comment. The part of the discussion was revised with references.
Before: There are many different opinions among researchers regarding the number of thermocycling treatments, but considering that an orthodontic treatment is usually carried out over a period of one year, the number of cycles was set to 10,000 in this experiment, which corresponds to the usage period of about one year in an oral cavity
After: There are many different opinions among researchers regarding the number of thermocycling treatments. In the standard ISO 11405, it is recommended that 500 thermal cycles in water bath at temperature of 5 and 55 °C, dwell time of 20 s be made [25]. However, it has been suggested that such a recommendation may not sufficient to reproduce the wear in an oral cavity, which led the prior researchers to increase the number of cycles [23, 26, 27]. According to Gale and Darvell [27], the 10,000 cycles of thermocycling corresponds to the usage period about one year in an oral cavity. The prior tests of dental bonding agent were usually performed after 5,000 or 10,000 cycles of thermocycling even if the composite materials retain more than five years in an oral cavity [14, 15, 22]. In this study, considering the period of orthodontic treatment, the number of cycles was set to 10,000.
Point 25. Lines 245-246: The authors must discuss why an increase in wettability is better for adhesion
Response 25: Thank you for your comment. The part of the manuscript was revised.
Before: The decrease in contact angle from 64.99 degree to 46.52 degree implies that the wettability increased. Based on these results, it can be said that by sandblasting the Y-TZP surface, a foundation was laid for effective resin bonding in a subsequent process
After: The decrease in contact angle from 64.99 degree to 46.52 degree implies that the wettability increased, therefore, it allows the polymer of the resin to flow into the Y-TZP surface [38]. Based on these results, it can be said that by sandblasting the Y-TZP surface, a foundation was laid for effective resin bonding in a subsequent process.
Point 26. The discussion should be improved, focusing more on the experiments performed and the obtained results. More detailed integration of results should be made.
Response 26: Thank you for your comment. The discussion of the manuscript was revised extensively by following your recommendation. Please refer to the attached file for revised discussion.
Point 27. References must be revised, and more actual references should be added to the manuscript
Response 27: Thank you for your comment. The references of the manuscript were revised by following your recommendation. Please refer to the attached file for revised reference.
Reviewer 3 Report
Introduction:
I would advise rewriting the hypotheses, using negation is confusing
I would recommend using the term "roughness" rather than "ruggedness"
Materials and Methods:
check spelling mistakes
no dimensions of zirconia samples were given
how large was the bonding area?
was the surface of the enamel sample flat on the whole bonding area? How was it obtained?
how long was the enamel etching?
Please, explain the "manufacturers' instructions" and "recommendations" for bonding agents used in the study
Please, describe in more detail the orthodontic brackets used in the study
section 2.2 description of study groups is confusing, I would suggest rearranging Table 1
I would also suggest renaming the study groups to clarify the study design, e.g. instead of "C", "P" as primer or "CP"; also "SB" can be confused with "SBS", "S" for sandblasting confused with "S", the name of the study group
In Table 2 the abbreviation for Zirconia is "MZ" while in the text authors mention "Y-TZP"; Check the abbrev. for Transbond XT adhesive primer throughout the text
Please, explain the sentence in lines 10-11 "The primer- and adhesive-applied surfaces were dried by enough blowing..."
Results:
Please explain the bracket's microstructure observations; in line 56 "The strength, color, height, and map of the bracket were examined by analyzing the topography" What is the map of the bracket? How was the strength or color of the bracket examined using topography anayzer?
Please, stick to the names of the sections mentioned in the Materials and methods: "Surface charateristics of Y-TZP" vs. "Surface roughness of Y-TZP" and "Contact angle on Y-TZP"
Discussion:
Line 216-218: how adding one more step, that is 10-MDP-containing primer, to the bonding procedure can simplify it? Also, in group S the Transbond XT adhesive primer was exchanged for Clearfil S3 bond (10-MDP-containing adhesive) so the number of steps remained the same.
What did the Authors mean by saying: "In this experiment, 50 μm alumina particles were sprayed in the same manner onto every Y-TZP sample.."?
Instead of "spraying the alumina on Y-TZP surface", it should be "sandblasting" or "alumina blasting"
Compare the hypotheses of the study and text in lines 255-260. Did you validate hypotheses based on SBS results before or after thermocycling?
Please, explain the results in group C (or CT); that study group used two primers: Clearfil ceramic primer (10-MDP-containing primer) and Transbond XT adhesive primer.
Conclusions:
Pleas, revise the conclusions:
in line 250 it is stated that ".. the bonding strength of 6–8 MPa required for bonding between a tooth and a bracket", and later in line 256-259: "However, after the thermocycling treatment, since the SBS decreased significantly to 4.99 and 4.31 MPa in the 10-MDP containing adhesive applied groups ST and CST, respectively, the bonding strength generally required for the bracket bonding was not satisfied."
Yet, in conclusions the authors stated that "When the 10-MDP-containing adhesive was used only, acceptable SBS was obtained" meaning 4.99 MPa obtained in group ST seems to be acceptable SBS.
Author Response
Point 1. I would advise rewriting the hypotheses, using negation is confusing
Response 1: Thank you for your comment. The negative null hypothesis was used since the long term stability of hydrophilic universal adhesive was unclear when used in bracket bonding.
Before: (1) Applying 10-MDP-containing adhesive to the sandblasted Y-TZP surface does not maintain the stable SBS of resin and Y-TZP; (2) Applying 10-MDP-containing primer to the sandblasted Y-TZP surface does not maintain the stable SBS of resin and Y-TZP.
After: The null hypotheses tested were as follows: Applying 10-MDP-containing universal adhesive to the sandblasted zirconia surface did not increased SBS between the ceramic bracket and zirconia.
Point 2. I would recommend using the term "roughness" rather than "ruggedness"
Response 2: Thank you for your comment. The “ruggedness” was revised to “roughness” throgh out the whole manuscript.
Materials and Methods:
Point 3. check spelling mistakes
Response 3: Thank you for your comment. The spelling mistakes was revised thoroughly.
Point 4. no dimensions of zirconia samples were given
Response 4: Thank you for your comment. The dimensions of zirconia samples were added in the manuscript.
Before: A total of 60 rectangular monolithic zirconia specimens were prepared from a green-stage Y-TZP block (LAVA Plus, 3M ESPE, USA) that was sintered according to the manufacturer’s instructions.
After: Sixty cuboidal Y-TZP specimens in 15 (width) × 15 (height) × 3 (thickness) mm were prepared from a green stage block (LAVA Plus, 3M ESPE, St. Paul, MN, USA) and sintered according to the manufacturer’s instructions.
Point 5. how large was the bonding area?
Response 5: Thank you for your comment. The information of bonding area was same as the area of the bracket base. The dimension (12.24 mm2) was added.
Before: A maxillary central incisor ceramic bracket (Perfect Clear II, Hubit, Korea) was used with orthodontic resin paste (XTa; Transbond XT adhesive paste, 3M Unitek, USA).
After: A ceramic bracket (Perfect Clear II, Hubit, Korea) for maxillary central incisor (bonding surface area of 12.24 mm2) was used.
Point 6. was the surface of the enamel sample flat on the whole bonding area? How was it obtained?
Response 6: Thank you for your comment. We used the maxillary incisors as it is without grinding to reproduce of natural teeth’s bonding when a patient is under an orthodontic treatment.
Point 7. how long was the enamel etching?
Response 7: The enamel is etched for 20 s following the manufacturers' instructions. The part of manuscript was revised.
Before: It was cleaned with distilled water for 2 min by using an ultrasonic machine and then dried, and then the labial surfaces were treated by using 37% phosphoric acid etchant (Scotchbond Universal Etchant, 3M ESPE, USA) according to the manufacturers’ instructions.
After: The excessive moisture was dried with air by using three-way syringe. The enamel was etched with 37% phosphoric acid (Scotchbond Universal Etchant, 3M ESPE, USA) for 20 s and then washed for 20 s using water, followed by drying with gentle air from the three-way syringe according to the manufacturers’ instructions.
Point 8. Please, explain the "manufacturers' instructions" and "recommendations" for bonding agents used in the study
Response 8: Thank you for your comment. The part of the materials and methods was revised by following your recommendation.
Before:
2.1. Specimen Preparation
2.1.2. Tooth
Extracted human teeth were immersed in a 0.1% thymol solution for a week and stored at 4 °C. Prior to the experiments, the teeth had been stored for less than 3 months, and they were used for the study with the approval of the School of Dentistry, Seoul National University Institutional Review Board (No. S-D20150041). After separating the dental root and crown of each extracted maxillary central incisor (n = 20), the crown was fixed to the acrylic resin so that the entire labial surface of the crown would be exposed and parallel to the base. It was cleaned with distilled water for 2 min by using an ultrasonic machine and then dried, and then the labial surfaces were treated by using 37% phosphoric acid etchant (Scotchbond Universal Etchant, 3M ESPE, USA) according to the manufacturers’ instructions.
As a control group (CON), the labial surface of right maxillary central incisors (n = 20) was treated with orthodontic primer (XTp; Transbond XT adhesive primer, 3M Unitek, USA) according to the manufacturers’ instructions.
2.2. Surface treatment and Bracket Bonding
The sandblasted Y-TZP samples were divided into three groups (C, S, and CS) according to the 10-MDP-containing primer and adhesive (n = 20) used. The Clearfil ceramic primer (CP; Kuraray, Japan) as a 10-MDP-containing primer, Clearfil S3 bond (SB; Kuraray, Japan) as a 10-MDP-containing adhesive, and XTp as a general orthodontic primer were used according to the manufacturers’ recommendations. For group C samples, CP was first applied on the Y-TZP surface, followed by the application of XTp. For group S samples, only SB was applied and light-cured. For group CS samples, SB was applied after CP was applied to the Y-TZP surface. The primer- and adhesive-applied surfaces were dried by enough blowing with a three-way air syringe in every application stage.
A maxillary central incisor ceramic bracket (Perfect Clear II, Hubit, Korea) was used with orthodontic resin paste (XTa; Transbond XT adhesive paste, 3M Unitek, USA). After an appropriate amount of XTa was applied to the bracket base, the bracket was placed under gentle pressure until the margin of the bracket base reached the Y-TZP surface. Excessive resin was removed with a resin applicator, and the bracket was cured with a light-emitting diode (LED) curing unit (Elipar Free Light 2, 3M ESPE, USA) at each margin for 10 s (total curing period: 40 s). The overall flow according to the bonding step is shown in Table 1. The components of materials used in this experiment are shown in Table 2.
After:
2.2. Bonding of bracket and shear bond strength test
2.2.1. Experimental design
The experimental design required 60 Y-TZP for three different experimental groups and 20 enamel specimens for a control group. Each group was divided into two subgroups (n=10), according to whether the aging treatment was performed. Figure 1 demonstrates the flow chart of this study, table 1 shows the specifications of the materials used in this study.
Group CON. Sound human maxillary central incisors (n=20) extracted for periodontal reasons were used under a protocol approved by an Institutional Review Board (No. S-D20150041). Maxillary central incisors (n = 20) which had been stored in a 0.1% thymol solution at 4 °C for less than 3 months after the extraction, were used. The roots were removed at the cemento-enamel junction using a diamond disk, the crown was fixed to the acrylic resin so that the entire labial surface of the crown would be exposed and parallel to the base. It was cleaned in distilled water with ultrasonic vibration for 2 min. The excessive moisture was dried with air by using three-way syringe. The enamel was etched with 37% phosphoric acid (Scotchbond Universal Etchant, 3M ESPE, USA) for 20 s and then washed for 20 s using water, followed by drying with gentle air from the three-way syringe according to the manufacturers’ instructions. Thereafter, an orthodontic primer (Transbond XT adhesive primer, 3M Unitek, USA) was applied to the enamel in a thin layer and the excessive orthodontic primer was removed by air blowing using three-way syringe according to the manufacturers’ instructions.
Group C. The sandblasted Y-TZP was treated with a ceramic primer (Clearfil ceramic primer, Kuraray, Japan) in a thin layer and dried thoroughly with gentle air of three-way syringe, followed by orthodontic primer application.
Group U. The sandblasted Y-TZP was treated with a universal adhesive (Clearfil S3 bond, Kuraray, Japan). The adhesive was applied in a thin layer, left for 20 s, dried for 5 s with the air of three-way syringe, and light-cured for 10 s.
Group CU. The sandblasted Y-TZP was treated with a ceramic primer, followed by application of a universal adhesive, then light-cured.
Point 9. Please, describe in more detail the orthodontic brackets used in the study
Response 9: Thank you for your comment. The information about the bracket was given in the manuscript.
After:
2.2.2. Bonding of bracket to specimen
A ceramic bracket (Perfect Clear II, Hubit, Korea) for maxillary central incisor (bonding surface area of 12.24 mm2) was used. The ceramic bracket used in this study was made of monocrystalline sapphire with retentive beads spreading at the center of the bracket base.
Point 10. section 2.2 description of study groups is confusing, I would suggest rearranging Table 1
Response 10: Thank you for your comment. The Table 1 of manuscript was deleted in the revised manuscript. Instead, figure 1 was inserted to depict the flow chart of the present study.
Before: Table 1. Experimental plan of this study.
Step 1. Pre-treatment | E | S | S | S | E | S | S | S |
Step 2. Primer | – | CP | – | CP | – | CP | – | CP |
Step 3. Adhesive | XTp | XTp | SB | SB | XTp | XTp | SB | SB |
Bracket bonding | XTa | XTa | XTa | XTa | XTa | XTa | XTa | XTa |
Thermocycling | N | N | N | N | Y | Y | Y | Y |
Group | CON | C | S | CS | CONT | CT | ST | CST |
After: Figure1.
Point 11. I would also suggest renaming the study groups to clarify the study design, e.g. instead of "C", "P" as primer or "CP"; also "SB" can be confused with "SBS", "S" for sandblasting confused with "S", the name of the study group
Response 11: Thank you for your comment. The name of group was revised by following your recommendation.
Before | After |
CON | CON0 |
CONT | CON1 |
C | CP0 |
CT | CP1 |
S | U0 |
ST | U1 |
CS | CU0 |
CST | CU1 |
Point 12. In Table 2 the abbreviation for Zirconia is "MZ" while in the text authors mention "Y-TZP"; Check the abbrev. for Transbond XT adhesive primer throughout the text
Response 12: Thank you for your comment. The part of the Table (abbreviation) was removed.
Point 13. Please, explain the sentence in lines 10-11 "The primer- and adhesive-applied surfaces were dried by enough blowing..."
Response 13: Thank you for your comment. This information was deleted and the details of the bonding step was added instead.
Before: The primer- and adhesive-applied surfaces were dried by enough blowing with a three-way air syringe in every application stage.
After: Group CON. Sound human maxillary central incisors (n=20) extracted for periodontal reasons were used under a protocol approved by an Institutional Review Board (No. S-D20150041). Maxillary central incisors (n = 20) which had been stored in a 0.1% thymol solution at 4 °C for less than 3 months after the extraction, were used. The roots were removed at the cemento-enamel junction using a diamond disk, the crown was fixed to the acrylic resin so that the entire labial surface of the crown would be exposed and parallel to the base. It was cleaned in distilled water with ultrasonic vibration for 2 min. The excessive moisture was dried with air by using three-way syringe. The enamel was etched with 37% phosphoric acid (Scotchbond Universal Etchant, 3M ESPE, USA) for 20 s and then washed for 20 s using water, followed by drying with gentle air from the three-way syringe according to the manufacturers’ instructions. Thereafter, an orthodontic primer (Transbond XT adhesive primer, 3M Unitek, USA) was applied to the enamel in a thin layer and the excessive orthodontic primer was removed by air blowing using three-way syringe according to the manufacturers’ instructions.
Group C. The sandblasted Y-TZP was treated with a ceramic primer (Clearfil ceramic primer, Kuraray, Japan) in a thin layer and dried thoroughly with gentle air of three-way syringe, followed by orthodontic primer application.
Group U. The sandblasted Y-TZP was treated with a universal adhesive (Clearfil S3 bond, Kuraray, Japan). The adhesive was applied in a thin layer, left for 20 s, dried for 5 s with the air of three-way syringe, and light-cured for 10 s.
Group CU. The sandblasted Y-TZP was treated with a ceramic primer, followed by application of a universal adhesive, then light-cured.
Results:
Point 14. Please explain the bracket's microstructure observations; in line 56 "The strength, color, height, and map of the bracket were examined by analyzing the topography" What is the map of the bracket? How was the strength or color of the bracket examined using topography anayzer?
Response 14: Thank you for your comment. In the revised manuscript, observation of bracket surface was excluded because it was considered less relevant to the present study.
Point 15. Please, stick to the names of the sections mentioned in the Materials and methods: "Surface charateristics of Y-TZP" vs. "Surface roughness of Y-TZP" and "Contact angle on Y-TZP"
Response 15: Thank you for your comment. The section of the manuscript was revised.
Before:
2. Materials and methods
2.1. Specimen Preparation
2.1.1. Zircnoia
2.1.2. Tooth
2.2. Surface treatment and Bracket Bonding
2.3. Observation of Microstructure of bracket
2.4. Surface charateristics of Y-TZP
2.5. Shear Bond Testing and Observation of Failure Mode
3. Result
3.1. Microstructure of bracket
3.2. Surface Roughness of Y-TZP
3.3. Contact Angle on Y-TZP
3.4. Shear Bond Strength and Failure Mode
After:
2. Materials and Methods
2.1. Preparation of Y-TZP
2.2. Observation of surface characteristics of Y-TZP
2.2.1. Field emission-scanning electron microscopic examination
2.2.2. Measurement of surface roughness
2.2.3. Measurement of contact angle
2.3. Bonding of bracket and shear bond strength test
2.3.1. Experimental design and bonding of the bracket to Y-TZP
2.3.2. Shear bond strength test
3. Results
3.1. Surface Characteristics of Y-TZP
3.1.1. Field emission-scanning electron microscopic examination
3.1.2. Measurement of surface roughness
3.1.3. Measurement of contact angle
3.2. Shear bond strength and failure mode
Discussion:
Point 16. Line 216-218: how adding one more step, that is 10-MDP-containing primer, to the bonding procedure can simplify it? Also, in group S the Transbond XT adhesive primer was exchanged for Clearfil S3 bond (10-MDP-containing adhesive) so the number of steps remained the same.
Response 16: Thank you for your comment. As you mentioned, 10-MDP-containing primer cannot simplify the bonding procedure.
For group U, a universal adhesive (Clearfil S3 bond) was applied to the sandblasted zirconia surface, then an orthodontic resin was applied. On the other hand, for group CP, ceramic primer was applied, followed by an orthodontic primer, then the orthodontic resin was applied. For group CU, ceramic primer was applied followed by the universal adhesive, then the orthodontic resin was applied. The procedure was depicted in figure 1 in the revised manuscript.
Before: This study evaluated the effectiveness of applying the 10-MDP-containing primer and 10-MDP-containing adhesive on sandblasted Y-TZP samples in order to simplify the steps of bonding between Y-TZP and the bracket.
After: Please refer to the Figure 1 about the number of steps and the change in the groups.
Figure1.
Point 17. What did the Authors mean by saying: "In this experiment, 50 μm alumina particles were sprayed in the same manner onto every Y-TZP sample.." Instead of "spraying the alumina on Y-TZP surface", it should be "sandblasting" or "alumina blasting"
Response 17: Thank you for your comment. The part of the manuscript was revised by following your recommendation.
Before: In this experiment, 50 μm alumina particles were sprayed in the same manner onto every Y-TZP specimen to only evaluate the effect of the 10-MDP-containing agents.
After: In this experiment, 50 μm alumina particles were sandblasted in the same manner onto every Y-TZP specimen to only evaluate the effect of the 10-MDP-containing agents.
Point 18. Compare the hypotheses of the study and text in lines 255-260. Did you validate hypotheses based on SBS results before or after thermocycling?
Response 18: Thank you for your comment. The null hypotheses and validation were revised.
Before (Hypotheses): (1) Applying 10-MDP-containing adhesive to the sandblasted Y-TZP surface does not maintain the stable SBS of resin and Y-TZP; (2) Applying 10-MDP-containing primer to the sandblasted Y-TZP surface does not maintain the stable SBS of resin and Y-TZP.
Before (validation): There was no significant difference in SBS before the thermocycling treatment between group C (9.78 ± 1.94 MPa), group S (9.86 ± 1.33 MPa), group CS (9.16 ± 0.78 MPa), and group CON (9.59 ± 1.77 MPa), and the bonding strengths generally required for the bonding of bracket were satisfied. The bonding steps can be simplified if the 10-MDP-containing adhesive is applied independently on the Y-TZP surface. Therefore, hypothesis 1 and 2 could not be validated. However, after the thermocycling treatment, since the SBS decreased significantly to 4.99 and 4.31 MPa in the 10-MDP-containing adhesive applied groups ST and CST, respectively, the bonding strength generally required for the bracket bonding was not satisfied. In contrast, group CT with 10-MDP-containing primer applied to zirconia retained stable SBS after thermocycling.
After (Hypotheses): The null hypotheses tested were as follows: Applying 10-MDP-containing universal adhesive to the sandblasted zirconia surface did not increased SBS between the ceramic bracket and zirconia.
After (Validation): There was no significant difference in SBS between groups without the thermocycling between group CP0 (9.78 ± 1.94 MPa), group U0 (9.86 ± 1.33 MPa), group CU0 (9.16 ± 0.78 MPa), and group CON0 (9.59 ± 1.77 MPa), and the bond strengths required for the bonding of bracket were satisfied. However, after the thermocycling, the SBS decreased significantly to 4.99 and 4.31 MPa in the groups U1 and CU1, respectively, which had been subjected to application of the universal adhesive. For groups U1 and CU1 the bonding strength generally required for the bracket bonding was not satisfied. In contrast, group CP1, with 10-MDP-containing ceramic primer applied to zirconia, retained stable SBS after thermocycling (8.16 ± 1.78 MPa). The group CP1 demonstrated the highest SBS, and there was no significant difference between the other three groups (CON1, U1, and CU1). Therefore, null hypothesis could be validated.
Point 19. Please, explain the results in group C (or CT); that study group used two primers: Clearfil ceramic primer (10-MDP-containing primer) and Transbond XT adhesive primer.
Response 19: Thank you for your comment. There was no significant difference in SBS before the thermocycling treatment between group C (9.78 ± 1.94 MPa), group S (9.86 ± 1.33 MPa), group CS (9.16 ± 0.78 MPa), and group CON (9.59 ± 1.77 MPa), and the bonding strengths generally required for the bonding of bracket were satisfied, group CT with 10-MDP-containing primer applied to zirconia retained stable SBS after thermocycling (8.16 ± 1.78 MPa)./ For group C with the 10- MDP-containing primer applied to the sandblasted Y-TZP, the SBS was 9.8 ± 1.8 MPa, and it was slightly reduced to 8.1 ± 1.6 MPa after thermocycling (group CT), which does not represent a significant difference.
Conclusions:
Point 20. Please, revise the conclusions:
in line 250 it is stated that ".. the bonding strength of 6–8 MPa required for bonding between a tooth and a bracket", and later in line 256-259: "However, after the thermocycling treatment, since the SBS decreased significantly to 4.99 and 4.31 MPa in the 10-MDP containing adhesive applied groups ST and CST, respectively, the bonding strength generally required for the bracket bonding was not satisfied."
Yet, in conclusions the authors stated that "When the 10-MDP-containing adhesive was used only, acceptable SBS was obtained" meaning 4.99 MPa obtained in group ST seems to be acceptable SBS.
Response 20: Thank you for your comment. SBS of 4.99 MPa obtained in group U1 does not reach the acceptable SBS, therefore, the part of the manuscript was revised by following your recommendation.
Before:
1. When the 10-MDP-containing adhesive was used only, acceptable SBS was obtained while simplifying the bonding step.
2. The long-term stable SBS values was obtained when the 10-MDP-containing primer followed by Transbond XT adhesive primer was applied to Y-TZP.
After: Within the limitations of this study, the null hypothesis was accepted. Based on the results, when bonding ceramic brackets to dental zirconia surface, we can conclude that ceramic primer used with an orthodontic primer, rather than using an universal adhesive is recommended.
Round 2
Reviewer 1 Report
Please see the minor corrections along the text.

Author Response
Point 1: You should write which parameters are evaluated with reference to their meaning. You should therefore delete the last sentence where you describe the mean of the acronym of the parameters
Response 1: Thank you for your comment. The manuscript was revised by following your recommendation.
Before: The roughness (Ra, Rq, Rz, Rsk) of the surface before and after sandblasting were determined by using a 3D surface confocal laser scanning microscope (CLSM; LSM 800-MAT, Carl Zeiss MicroImaging GmbH, Germany). The roughness parameters mean: Ra, average surface roughness; Rq, root mean square roughness; Rz, mean roughness depth; and Rsk, roughness skewness
After: The surface was observed under a 3D surface confocal laser scanning microscope (CLSM; LSM 800-MAT, Carl Zeiss MicroImaging GmbH, Germany). The roughness parameters evaluated were average surface roughness (Ra), root mean square roughness ( Rq), roughness depth (Rz), and roughness skewness (Rsk).
Point 2: After bracket bonding, all the samples were kept in a 37 °C and relative humidity 100% incubator for 24 h. Half of the samples were then randomly selected from each group (CON0, CP0, U0, and 151 CU0), and the shear bond strength of these samples was measured with a universal testing machine 152 (Instron 8848, Instron, USA) at a crosshead speed of 0.5 mm/min. 153
How?
Response 2: Thank you for your comment. The manuscript was revised by following your recommendation.
Before: After bracket bonding, all the samples were kept in a 37 °C and relative humidity 100% incubator for 24 h. Half of the samples were then randomly selected from each group (CON0, CP0, U0, and CU0), and the shear bond strength of these samples was measured with a universal testing machine (Instron 8848, Instron, USA) at a crosshead speed of 0.5 mm/min. The remaining samples (CON1, CP1, U1, and CU1) were subjected to the aging process by thermocycling for 10,000 cycles at 5 and 55 °C. The dwelling time in water was 30 s, and the transfer time was 20 s.
After: After bracket bonding, all the samples (n = 80) were kept in a 37 °C and relative humidity 100% incubator for 24 h. SBS of forty samples, ten per each group, tested with a universal testing machine (Instron 8848, Instron, USA) at a crosshead speed of 0.5 mm/min (CON0, CP0, U0, and CU0). The other half were subjected to the aging process by thermocycling for 10,000 cycles at 5 and 55 °C (CON1, CP1, U1, and CU1). The dwelling time in water was 30 s, and the transfer time was 20 s. After thermocycling, SBS of the samples tested in the same way.
Point 3: The conclusions are better but can it be improved. The construction of sentences should be reviewed.
Response 3: Thank you for your comment. The manuscript was revised by following your recommendation.
Before: Within the limitations of this study, the null hypothesis was accepted. Based on the results, when bonding ceramic brackets to dental zirconia surface, we can conclude that ceramic primer used with an orthodontic primer, rather than using an universal adhesive is recommended.
After: In the present study, the long term stability of shear bond strength when 10-MDP-containing universal adhesive was used in the ceramic bracket bonding on dental zirconia.
In ceramic bracket bonding, the use of a universal adhesive showed a clinically acceptable range of the shear bond strength after 24 h. After artificial aging process, the shear bond strength of the universal adhesive was more affected when compared with using a ceramic primer and an orthodontic primer.
In clinical aspects, it can be concluded that:
When bonding ceramic brackets to a dental zirconia, the application of the ceramic primer on the zirconia followed by an orthodontic primer is recommended rather than using a universal adhesive.
Reviewer 3 Report
Thank you for revising your paper according to previous suggestions. Please, include the following corrections:
Abstract:
Please, correct the names of the study groups: line 25-26 "The shear bond strength (SBS) test was tested before (CON1, CP1, U1, CU1) and after the artificial aging (CON0, CP0, U0, CU0)." whereas in Fig 1 they are coded the opposite. Also remove "was" from the mentioned sentence.
line 28: "After the aging process, SBS were decreased in all groups." Please, remove "were" from the sentence.
Introduction:
line 46-48 the sentence misses a verb
line 59: please mention, which acid-etching you are referring to: orthophosphoric or hydrofluoric. Since the control in the study is bonding to enamel, the phrase "acid-etching" is misleading
line 77: Please, change to: "The null hypothesis tested was that applying 10-MDP-containing universal adhesive to the sandblasted zirconia surface do not increase SBS between the ceramic bracket and zirconia".
Materials and Methods:
lilne 82: Were the presented dimensions obtain before or after sintering?
line 107: remove "of"
line 108: Did the authors consider creating study group where on sandblasted Y-TZP surface an orthodontic primer is applied? Please, explain.
line 130: Please, correct the code for study group: "C" to "CP"
line 169: change to "x5000"
line 175: remove "which was"
line 193: change to: "decreased the SBS exept for the group CP.."
line 197: remove "were"
Discussion:
line 223: change to "may not be sufficient to reproduce.." or "may not sufficiently reproduce.. "
line 233: Please explain: "and it is the usual method of surface treatment in dentistry.."?
line 247-249: rearrange the sentence as follows: "However, if alumina particles are sprayed on Y-TZP, the surface area as well as the surface energy of Y-TZP will increase, improving the wettability, thereby contributing to the increased bonding strength."
line 250: remove "of"
line 251: Please, explain the results of other roughness parameters, e.g. Rz increased from 0.96 to 4.36 μm
line 264: change to: "where the universal adhesive was applied"
line 288: change to: "universal adhesive"
line 292, 294: change to "Y-TZP"
line 292-298: Regarding the results of resin remnants on Y-TZP surface: Please, explain the ARI scores. ARI score of 3 indicate cohesive failure within the composite resin and clearly do not indicate the SBS between zirconia and composite resin. Therefore no statistical difference between SBS of groups CP, U, CU. Are the obtained SBS results reliable? or perhaps the shear bond test design should be changed?
Conclusions:
Conclusions should correspond with the aim of the study.
Author Response
Abstract:
Point 1: Please, correct the names of the study groups: line 25-26 "The shear bond strength (SBS) test was tested before (CON1, CP1, U1, CU1) and after the artificial aging (CON0, CP0, U0, CU0)." whereas in Fig 1 they are coded the opposite. Also remove "was" from the mentioned sentence.
Response 1: Thank you for your comment. The manuscript was revised by following your recommendation.
Before: The shear bond strength (SBS) test was tested before (CON1, CP1, U1, CU1) and after the artificial aging (CON0, CP0, U0, CU0).
After: The shear bond strength (SBS) tested before (CON0, CP0, U0, CU0) and after the artificial aging (CON1, CP1, U1, CU1).
Point 2: line 28: "After the aging process, SBS were decreased in all groups." Please, remove "were" from the sentence.
Response 2: Thank you for your comment. The manuscript was revised by following your recommendation.
Before: After the aging process, SBS were decreased in all groups
After: After the aging process, SBS decreased in all groups
Introduction:
Point 3: line 46-48 the sentence misses a verb
Response 3: Thank you for your comment. The manuscript was revised by following your recommendation.
Before: The chipping or crack of the porcelain attributed to the poor wetting of the veneering, the firing shrinkage of the porcelain, and the difference in the thermal expansion coefficient between the zirconia and the porcelain
After: The chipping or crack of the porcelain is attributed to the poor wetting of the veneering, the firing shrinkage of the porcelain, and the difference in the thermal expansion coefficient between the zirconia and the porcelain.
Point 4: line 59: please mention, which acid-etching you are referring to: orthophosphoric or hydrofluoric. Since the control in the study is bonding to enamel, the phrase "acid-etching" is misleading
Response 4: Thank you for your comment. The manuscript was revised by following your recommendation.
Before: When the tooth is restored with MZ, bonding brackets on the zirconia is a challenging process since the zirconia surface, which has no silica phase, cannot be bonded with composite resin through conventional acid-etching and silane treatment
After: When the tooth is restored with MZ, bonding brackets on the zirconia is a challenging process since the zirconia surface, which has no silica phase, cannot be bonded with composite resin through conventional hydrofluoric acid-etching and silane treatment
Point 5: line 77: Please, change to: "The null hypothesis tested was that applying 10-MDP-containing universal adhesive to the sandblasted zirconia surface do not increase SBS between the ceramic bracket and zirconia".
Response 5: Thank you for your comment. The manuscript was revised.
Before: The null hypotheses tested were as follows: Applying 10-MDP-containing universal adhesive to the sandblasted zirconia surface did not increased SBS between the ceramic bracket and zirconia.
After: The aim of this study was to evaluate the long term stability of shear bond strength (SBS) when 10-MDP-containing universal adhesive was used in the ceramic bracket bonding on dental zirconia.
Materials and Methods:
Point 6: lilne 82: Were the presented dimensions obtain before or after sintering?
Response 6: Thank you for your comment. The manuscript was revised by following your recommendation.
Before: Sixty cuboidal Y-TZP specimens in 15 (width) × 15 (height) × 3 (thickness) mm were prepared from a green-stage block (LAVA Plus, 3M ESPE, St. Paul, MN, USA) and sintered according to the manufacturer’s instructions.
After: Sixty cuboidal Y-TZP specimens in 15 (width) × 15 (height) × 3 (thickness) mm were prepared from a green-stage block (LAVA Plus, 3M ESPE, St. Paul, MN, USA), and then the specimens were sintered according to the manufacturer’s instructions.
Point 7: line 107: remove "of"
Response 7: Thank you for your comment. The manuscript was revised by following your recommendation.
Before: Experimental design and bonding of the bracket to Y-TZP
After : Experimental design and bonding the bracket to Y-TZP
Point 8: line 108: Did the authors consider creating study group where on sandblasted Y-TZP surface an orthodontic primer is applied? Please, explain.
Response 8:
Yes, we also evaluated direct bonding with an orthodontic primer which was not shown in this study and had the average SBS of 5.71 ± 0.48 MPa before thermocycling.
As listed in the table1 (materials used in the study), unlike the ceramic primer, the orthodontic adhesive primer mainly contains TEGDMA, Bis-GMA and to achieve the desired SBS of the brackets the use is only limited on the enamel surface. This was the reason why we only showed the result of the sandblasted zirconia surface treated with the ceramic primer. Another reason was to compare the use of an universal adhesive with the widely used zirconia bonding protocol which is an application of ceramic primer followed by an orthodontic primer.
Point 9: line 130: Please, correct the code for study group: "C" to "CP"
Response 9: Thank you for your comment. The manuscript was revised by following your recommendation.
Before: Group C. The sandblasted Y-TZP was treated with a ceramic primer (Clearfil ceramic primer, Kuraray, Japan) in a thin layer and dried thoroughly with gentle air of three-way syringe, followed by orthodontic primer application.
After: Group CP. The sandblasted Y-TZP was treated with a ceramic primer (Clearfil ceramic primer, Kuraray, Japan) in a thin layer and dried thoroughly with gentle air of three-way syringe, followed by orthodontic primer application.
Point 10: line 169: change to "x5000"
Response 10: Thank you for your comment. The manuscript was revised by following your recommendation.
Point 11: line 175: remove "which was"
Response 11: Thank you for your comment. The manuscript was revised by following your recommendation.
Before: When the polished surface of Y-TZP, which was not subjected to sandblasting,
After: When the polished surface of Y-TZP, not subjected to sandblasting,
Point 12: line 193: change to: "decreased the SBS except for the group CP.."
Response 12: Thank you for your comment. The manuscript was revised by following your recommendation.
Before: Thermocycling decreased the SBS except group CP
After: Thermocycling decreased the SBS except for the group CP
Point 13: line 197: remove "were"
Response 13: Thank you for your comment. The manuscript was revised by following your recommendation.
Discussion:
Point 14: line 223: change to "may not be sufficient to reproduce.." or "may not sufficiently reproduce.. "
Response 14: Thank you for your comment. The manuscript was revised by following your recommendation.
Before: However, it has been suggested that such a recommendation may not sufficient to reproduce the wear in an oral cavity
After: However, it has been suggested that such a recommendation may not be sufficient to reproduce the wear in an oral cavity
Point 15: line 233: Please explain: "and it is the usual method of surface treatment in dentistry.."?
Response 15: Thank you for your comment. The alumina blasting to bonding surface of the zirconia is common method for enhancement of bond strength between zirconia and resin cement.
Point 16: line 247-249: rearrange the sentence as follows: "However, if alumina particles are sprayed on Y-TZP, the surface area as well as the surface energy of Y-TZP will increase, improving the wettability, thereby contributing to the increased bonding strength."
Response 16: Thank you for your comment. The manuscript was revised by following your recommendation.
Before: However, if alumina particles are sprayed on Y-TZP, the surface area of Y-TZP will increase and the surface energy will also increase, thereby contributing to the increased bonding-strength by improving the wettability
After: However, if alumina particles are sandblasted on Y-TZP, the surface area as well as the surface energy of Y-TZP will increase, improving the wettability, thereby contributing to the increased bonding-strength
Point 17: line 250: remove "of"
Response 17: Thank you for your comment. The manuscript was revised by following your recommendation.
Point 18: line 251: Please, explain the results of other roughness parameters, e.g. Rz increased from 0.96 to 4.36 μm
Response 18: Thank you for your comment. The manuscript was revised by following your recommendation.
Before: The increase in surface roughness from 0.10 µm to 0.70 µm indicates an increase in surface area that can be available for chemical reactions
After: The increase in Ra from the average of 0.10 µm to 0.70 µm, together with increased Rq (from the average of 0.14 µm to 0.85 µm) and Rz (from the average of 0.96 µm to 4.36 µm) indicates an increase in surface area that can be available for chemical reactions. The decreased in Rsk from the average of 0.96 µm to -0.28 µm implies the roughness was uniformly increased when compared with polished surface with short multiple scratches. Through the surface treatment with sand blasting, the contact angle was decreased from 64.99 to 46.52 degrees, indicating an increased wettability, which allows the polymer of the resin to flow into the Y-TZP surface [38]. Based on these results, it can be said that by sandblasting the Y-TZP surface, a foundation was laid for effective resin bonding in a subsequent process.
Point 19: line 264: change to: "where the universal adhesive was applied"
Response 19: Thank you for your comment. The manuscript was revised by following your recommendation.
Before: However, after the thermocycling, the SBS decreased significantly to 4.99 and 4.31 MPa in the groups U1 and CU1, respectively, which had been subjected to application of the universal adhesive
After: However, after the thermocycling, the SBS decreased significantly to 4.99 and 4.31 MPa in the groups U1 and CU1, respectively, where the universal adhesive was applied
Point 20: line 288: change to: "universal adhesive"
Response 20: Thank you for your comment. The manuscript was revised by following your recommendation.
Before: a self-etch adhesive
After: an universal adhesive
Point 21: line 292, 294: change to "Y-TZP"
Response 21: Thank you for your comment. The manuscript was revised by following your recommendation.
Before: YTZP
After: Y-TZP
Point 22: line 292-298: Regarding the results of resin remnants on Y-TZP surface: Please, explain the ARI scores. ARI score of 3 indicate cohesive failure within the composite resin and clearly do not indicate the SBS between zirconia and composite resin. Therefore no statistical difference between SBS of groups CP, U, CU. Are the obtained SBS results reliable? or perhaps the shear bond test design should be changed?
Response 22: Thank you for your comment.
A bracket bonding failure in clinical situation is mainly occurred between the resin and brackets rather than between the tooth surface and resin during orthodontic treatment. But when the bracket is bonded to the zirconia, the bracket bonding failure is mainly occurred between zirconia and resin rather than between resin and bracket due to high mechanical properties of zirconia. In accordance with this clinical aspect, we focused on improving the bond strength between zirconia and resin, not between resin and bracket, and the experimental was designed the according to the general protocol that is used in other experiments 1,2.3. The shear bond strength was measured because the force contributing to the dislocation of the bracket in the oral cavity was closer to the shear bond stress than the tensile bond stress. As a result of this experiment, the shear bond strength (in case of non-thermocycling) similar to those of the control group (natural teeth) were observed in all experimental groups (zirconia), and all bonding failures occurred between the resin and the bracket. In another study of our team, which is under preparation for the publication, the primer treatment of the brackets definitely improved the bond strength between the brackets and the resin, and the failure at the zirconia and the resin interface was observed. However, the focus of this study was to find how an universal adhesive performs (U, CU) after aging process compared with a conventionally accepted zirconia bonding protocol (CP).
1 N.-H. Kim,, Y.-J. Kim, and D.-Y. Lee, Bond Strengths of Orthodontic Metal Brackets to Tribochemically Silica-coated Zirconia Surfaces Using Different 10-Methacryloyloxydecyl Dihydrogen Phosphate-containing Primers. The journal of adhesive dentistry, 2017.
2 J.-Y. Lee,, et al., Comparison of bond strengths of ceramic brackets bonded to zirconia surfaces using different zirconia primers and a universal adhesive. Restorative Dentistry & Endodontics, 2018.
3 S.-M. Byeon, M.-H. Lee, T.-S. Bae, Shear Bond Strength of Al 2 O 3 Sandblasted Y-TZP Ceramic to the Orthodontic Metal Bracket. Materials, 2007
Conclusions:
Point 23: Conclusions should correspond with the aim of the study.
Response 23: Thank you for your comment. The manuscript was revised by following your recommendation.
Before: Within the limitations of this study, the null hypothesis was accepted. Based on the results, when bonding ceramic brackets to dental zirconia surface, we can conclude that ceramic primer used with an orthodontic primer, rather than using an universal adhesive is recommended.
After: In the present study, the long term stability of shear bond strength when 10-MDP-containing universal adhesive was used in the ceramic bracket bonding on dental zirconia.
In ceramic bracket bonding, the use of a universal adhesive showed a clinically acceptable range of the shear bond strength after 24 h. After artificial aging process, the shear bond strength of the universal adhesive was more affected when compared with using a ceramic primer and an orthodontic primer.
In clinical aspects, it can be concluded that:
When bonding ceramic brackets to a dental zirconia, the application of the ceramic primer on the zirconia followed by an orthodontic primer is recommended rather than using a universal adhesive.